# Learning Permutation from Structure Without Supervision

**Ran Eisenberg** [1]  **Ofir Lindenbaum** [1]

## Abstract

Many learning problems require uncovering a hidden ordering that reveals structure in unordered data, such as monotonicity in sorting or spatial continuity in jigsaw reconstruction. In these settings, permutations can be learned as latent operators by optimizing objectives defined directly on the reordered output, often without access to ground-truth orderings. Differentiable relaxations such as Gumbel–Sinkhorn make this approach practical by approximating permutation matrices with doubly stochastic matrices. However, learning from structure without supervision induces a non-uniform uncertainty: some assignments become confident early, while others remain ambiguous. Existing methods control this process using a single global temperature, forcing all assignments to sharpen or diffuse simultaneously and leading to instability at scale. We introduce an entropy-adaptive formulation of Gumbel–Sinkhorn that locally modulates temperature based on assignment uncertainty. This allows confident assignments to discretize early while preserving exploration where uncertainty remains. Across sorting and jigsaw reconstruction tasks and in routing-style settings, adaptive entropy control improves training stability and final permutation quality relative to fixed-temperature baselines, particularly as problem size and assignment ambiguity increase.

## 1. Introduction

Many tasks involve data that are naturally unordered at observation time and can be addressed as recovering a hidden permutation. For example, sorting requires reordering elements into a monotonic sequence (Grover et al., 2019; Prillo & Eisenschlos, 2020; Cuturi et al., 2019; Blondel

---

[1]Faculty of Engineering, Bar-Ilan University, Ramat Gan, Israel. Correspondence to: Ran Eisenberg <eisenbr2@biu.ac.il>, Ofir Lindenbaum <ofir.lindenbaum@biu.ac.il>.

*Proceedings of the $43^{rd}$ International Conference on Machine Learning*, Seoul, South Korea. PMLR 306, 2026. Copyright 2026 by the author(s).

et al., 2020), while jigsaw reconstruction seeks a spatial arrangement that produces visual continuity (Santa Cruz et al., 2017; Mena et al., 2018); ranking, matching, and routing problems similarly reduce to finding an ordering that reveals latent structure (Emami & Ranka, 2018; Min et al., 2023). In these settings, structure emerges only after the data are correctly ordered, suggesting a natural unsupervised formulation when ground-truth permutations are unavailable. Rather than predicting permutations as labels, they can be treated as operators applied to data, with learning driven by whether the reordered output satisfies task-specific structural properties such as smoothness, monotonicity, or geometric coherence.

Recent advances in differentiable permutation learning make this operator-based view practically usable. Continuous relaxations approximate permutation matrices by doubly stochastic matrices and enable end-to-end optimization via Sinkhorn normalization and related operators (Santa Cruz et al., 2017; Mena et al., 2018; Cuturi, 2013). To encourage convergence toward discrete permutations, these methods rely on a temperature parameter that controls the entropy of the relaxation and is typically annealed during training (Mena et al., 2018; Emami & Ranka, 2018; Guo et al., 2024).

Despite their success on small or strongly constrained problems, such approaches become fragile as problem size and ambiguity increase. In realistic permutation learning tasks, different parts of the assignment resolve at different rates: some rows or columns become nearly deterministic early in training, while others remain ambiguous due to symmetry or weak structural cues. A single global temperature forces all assignments to sharpen or diffuse simultaneously, leading to an unavoidable trade-off between stability and discretization. Aggressive annealing can lock in early mistakes, while conservative annealing prevents large, ambiguous permutations from sharpening sufficiently. Section 3.3 isolates this trade-off in a block-ambiguous construction, showing why heterogeneous assignment uncertainty can make scalar temperature schedules difficult to tune. This stability–discretization tension closely mirrors known trade-offs in entropically regularized optimal transport (Chizat, 2024).

We address this limitation with an entropy-adaptive formula-

tion of Gumbel–Sinkhorn that locally modulates inverse temperature based on assignment uncertainty. Instead of a single global scalar, we introduce a row- and column-dependent inverse-temperature field derived from the entropy of the current soft permutation. This allows confident assignments to discretize early while preserving exploration in ambiguous regions, and is equivalent to locally rescaling assignment costs over the Birkhoff polytope (Section 3.4). The proposed formulation is a lightweight extension of the Gumbel–Sinkhorn family: it changes how temperature is parameterized, while leaving the relaxation, constraints, decoding procedure, and underlying convergence properties of Gumbel–Sinkhorn unchanged. Figure 1 provides an overview of the proposed entropy-adaptive inverse-temperature mechanism, illustrating how confident assignments discretize early while ambiguous regions remain diffuse.

We make the following contributions:

- We propose a unified framework for learning permutations from structure, treating them as latent operators and training them directly from task-specific structural losses on reordered data. This enables unsupervised learning without access to ground-truth permutations.

- We introduce an entropy-adaptive variant of Gumbel–Sinkhorn that replaces a single global temperature with a locally varying temperature field defined at the row and column level. Inverse temperature is reduced for assignments with high uncertainty, allowing confident matches to discretize early while preserving exploration elsewhere.

- We show that applying Sinkhorn normalization with an entrywise inverse-temperature field is equivalent to solving an entropically regularized assignment problem with locally rescaled costs, providing an optimal transport interpretation of adaptive temperature control over the Birkhoff polytope.

- Across sorting, jigsaw reconstruction, and unsupervised Travelling Salesman Problem (TSP), entropy-adaptive temperature control yields more reliable convergence and higher-quality permutations than fixed or globally annealed temperatures, particularly as the number of elements $n$ increases and assignment ambiguity becomes heterogeneous. Our code implementation is available through this link.

## 2. Related Work

Learning and reasoning over permutations has received attention in recent years, motivated by tasks in which structure is revealed only after elements are reordered. A common strategy is to represent permutations using continuous relaxations that enable gradient-based optimization while approximating discrete permutation matrices.

**Differentiable permutation relaxations.** Early work introduced differentiable layers that approximate permutation matrices using Sinkhorn normalization, enabling end-to-end learning of assignments within neural networks. Deep-PermNet employs unrolled Sinkhorn iterations to predict doubly stochastic matrices as soft permutations for visual reordering tasks (Santa Cruz et al., 2017). Mena et al. formalize this approach with Sinkhorn networks and introduce the Gumbel–Sinkhorn distribution, which combines Gumbel noise with entropic regularization to sample differentiable approximations of permutation matrices that converge to discrete permutations as temperature decreases (Mena et al., 2018).

These methods are closely related to entropy-regularized optimal transport, where Sinkhorn normalization arises as the solution to a regularized assignment problem over the Birkhoff polytope (Cuturi, 2013). More recent work explores alternative relaxation geometries for permutation learning. OT4P maps unconstrained parameters to the orthogonal group using a temperature-dependent transformation that concentrates mass near permutation matrices as temperature decreases (Guo et al., 2024). While differing in geometry, these approaches similarly rely on a global temperature parameter to control the sharpness of the relaxation.

**Stochastic optimization and perturbation-based relaxations.** Beyond deterministic relaxations, stochastic formulations enable learning with latent permutations through reparameterized sampling. Sinkhorn Policy Gradient (SPG) integrates a temperature-controlled Sinkhorn operator into reinforcement learning pipelines to optimize permutation-structured actions (Emami & Ranka, 2018). Perturbation-based relaxations such as the Gumbel–Softmax and Concrete distributions provide a general framework for differentiable sampling of discrete random variables (Jang et al., 2017; Maddison et al., 2017), of which Gumbel–Sinkhorn can be viewed as a structured, matrix-valued extension (Mena et al., 2018).

Recent theoretical work has clarified the close connection between entropy regularization and perturbation-based smoothing in differentiable subset selection and structured prediction (Sun & others, 2023). Our method builds directly on this perspective by making temperature adaptive and spatially varying, rather than uniform.

**Differentiable sorting, ranking, and selection.** A related line of work focuses on differentiable surrogates for sorting, ranking, and top-$k$ selection operators. NeuralSort introduces a continuous relaxation of the sorting operator that enables stochastic gradient estimation over permutations (Grover et al., 2019). SoftSort proposes a simple and

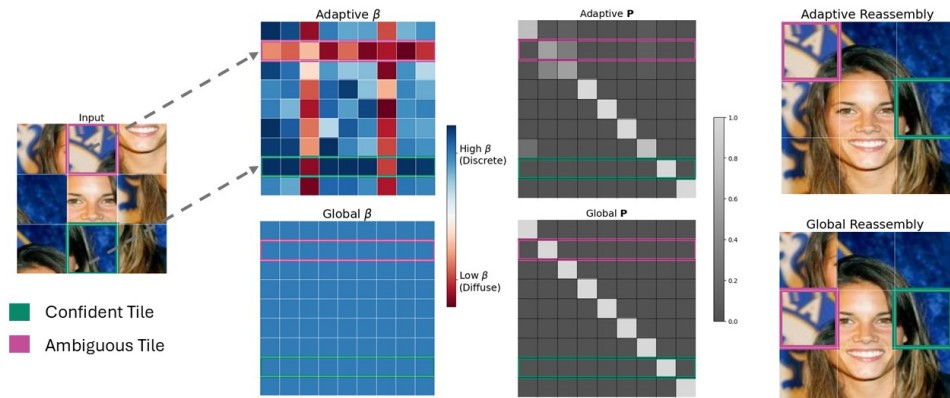

**Figure 1.** Overview of entropy-adaptive inverse-temperature control for learning permutations with Gumbel–Sinkhorn. Starting from a scrambled input image (left), the model predicts a soft permutation over tile positions. The adaptive inverse-temperature field $\boldsymbol{\beta}$ varies across rows and columns based on assignment uncertainty, assigning high $\beta$ to confident tiles while keeping ambiguous tiles diffuse. In contrast, a single global $\beta$ applies uniform sharpening to all assignments. The resulting soft permutation matrices $\boldsymbol{P}$. Adaptive temperature yields a mixture of sharp and soft rows, whereas global temperature forces all rows to discretize uniformly. Final reassemblies obtained by decoding the soft permutations. Adaptive temperature preserves flexibility in ambiguous regions while correctly fixing confident tiles, thereby improving reconstruction quality.

efficient relaxation of the `argsort` operator with favorable empirical behavior (Prillo & Eisenschlos, 2020). Other approaches connect sorting to optimal transport: Cuturi et al. formulate differentiable sorting and ranking as entropically regularized optimal transport problems solved via Sinkhorn iterations (Cuturi et al., 2019), while Blondel et al. propose fast differentiable sorting and ranking via projections onto the Permutahedron (Blondel et al., 2020).

Differentiable top-$k$ and subset selection methods further generalize these ideas beyond full permutations (Petersen et al., 2022). These methods are designed to approximate functional operators such as sorting or selection directly, rather than to learn a latent permutation matrix. In contrast, our focus is on learning permutations as latent operators optimized through downstream structural objectives.

**Annealing and entropy control in optimal transport.** Entropic regularization is typically controlled through a single global parameter. In optimal transport, annealed Sinkhorn schemes vary the regularization strength over iterations to trade off approximation accuracy and convergence behavior (Chizat, 2024). While such schedules improve numerical stability and solution quality, the regularization remains uniform across the transport plan. In contrast, our method introduces heterogeneous, uncertainty-aware entropy control by adapting temperature locally based on assignment uncertainty within the permutation matrix. Related work has also explored adaptive or locally varying regularization in optimal transport and matching problems. Our approach is conceptually related but differs in scope and mechanism: we introduce adaptive inverse temperature within the Gumbel–Sinkhorn relaxation for latent permutation learning, using row- and column-level assignment entropy as an uncertainty signal, and integrate it with stop-gradient control and down-

stream structural losses. This enables uncertainty-aware discretization in unsupervised permutation learning settings, rather than improving transport accuracy for a fixed matching objective.

**Unsupervised permutation learning for combinatorial optimization.** Unsupervised approaches to combinatorial optimization often rely on surrogate objectives combined with decoding procedures. Unsupervised Learning for Solving the Travelling Salesman Problem (UTSP) trains a graph neural network to predict a soft edge heat map optimized for short tours, followed by search-based decoding to obtain valid TSP solutions (Min et al., 2023). Subsequent work reformulates TSP explicitly as permutation learning using Gumbel–Sinkhorn relaxations and analyzes the role of entropy in solution quality (Min & Gomes, 2023; 2025). Our work is closest in spirit to these permutation-based approaches but differs in scope: we treat permutation learning as a general-purpose operator and focus on stabilizing learning via local entropy adaptation rather than relying on a single global temperature schedule or fixed structural templates.

**Unsupervised jigsaw reconstruction.** Jigsaw-style self-supervision provides a natural testbed for permutation learning, where structure emerges only after correct reordering. Differentiable permutation relaxations enable direct optimization of reconstruction losses on reordered inputs (Mena et al., 2018). Recent work explores alternative unsupervised formulations that decode permutations sequentially, for example, by tokenizing pieces and using autoregressive sequence-to-sequence models to predict placements (Elkin et al., 2025). Compared to such discrete sequential decoding approaches, our framework predicts a soft permutation matrix in one shot via adaptive Gumbel–Sinkhorn and applies

losses directly to the reordered output, making local entropy control central to scalability and stability.

## 3. Method

We consider permutation learning as a differentiable assignment problem. Given an unordered set of $n$ elements $\boldsymbol{X} \in \mathbb{R}^{n \times d}$, the goal is to learn a permutation that maps these elements to an ordered set of $n$ positions.

A (discrete) permutation is represented by a permutation matrix $\boldsymbol{\Pi} \in \{0, 1\}^{n \times n}$ satisfying $\boldsymbol{\Pi} \mathbf{1} = \mathbf{1}$ and $\boldsymbol{\Pi}^\top \mathbf{1} = \mathbf{1}$, where $\Pi_{ij} = 1$ indicates that input element $i$ is assigned to position $j$ and $\mathbf{1} \in \mathbb{R}^n$ is the vector of all ones. These matrices are in one-to-one correspondence with the symmetric group $S_n$. Applying a permutation reorders the input as $\hat{\boldsymbol{X}} = \boldsymbol{\Pi}^\top \boldsymbol{X}$.

To enable gradient-based optimization, we relax permutation matrices to the Birkhoff polytope and represent soft permutations by $\boldsymbol{P} \in \mathcal{B}_n$, where

$$\mathcal{B}_n = \left\{ \boldsymbol{P} \in \mathbb{R}_+^{n \times n} \mid \boldsymbol{P} \mathbf{1} = \mathbf{1}, \ \boldsymbol{P}^\top \mathbf{1} = \mathbf{1} \right\}$$

is the Birkhoff polytope. In this relaxed setting, $\hat{\boldsymbol{X}} = \boldsymbol{P}^\top \boldsymbol{X}$ defines a soft reordering.

Given the unordered set of elements $\boldsymbol{X}$, a neural network $f_\theta$ outputs a matrix of compatibility scores $f_\theta(\boldsymbol{X}) = \boldsymbol{S} \in \mathbb{R}^{n \times n}$, where $\theta$ denotes the model's parameters, and $S_{ij}$ denotes the compatibility score between input element $i$ and position $j$. Using $\boldsymbol{S}$, a soft permutation $\boldsymbol{P}$ is then sampled from a differentiable relaxation of the permutation distribution and optimized as a latent variable following (Mena et al., 2018). At a high level, the Gumbel-Sinkhorn method serves as the matrix-valued counterpart to Gumbel-Softmax. The Sinkhorn operator transforms logits into the Birkhoff polytope, while the temperature parameter determines whether the resulting doubly stochastic matrix is sparse or nearly discrete.

$$\boldsymbol{P} = \text{Sinkhorn}\Big(\boldsymbol{S} + \boldsymbol{g}\Big), g_{ij} \sim \text{Gumbel}(0, 1). \quad (1)$$

where $g_{ij}$ is i.i.d. Gumbel noise $g_{ij} \sim \text{Gumbel}(0, 1)$ added to the assignment logits; and $\text{Sinkhorn}(\cdot)$ is the Sinkhorn operator, which iteratively normalizes rows and columns of a matrix, as defined in (Adams & Zemel, 2011)[1]. Deterministic Sinkhorn alone is sufficient when one only needs a differentiable soft assignment for fixed logits, but it does not by itself sample from a latent distribution over permutations. The Gumbel perturbation supplies a reparameterized stochastic search over competing matchings, encourages exploration when several assignments have similar scores,

---

[1]See additional details in Appendix Q

---

**Algorithm 1** Entropy-adaptive Gumbel–Sinkhorn (inverse temperature)

---

**Require:** Scores $\boldsymbol{S} \in \mathbb{R}^{n \times n}$, base inverse temperature $\beta_0(t)$
**Ensure:** Soft permutation $\boldsymbol{P}$
1: $\boldsymbol{Q} \leftarrow \text{Sinkhorn}(\beta_0(t)\,\boldsymbol{S})$
2: Compute $H^{\text{row}}, H^{\text{col}}$ as in Eq. (3)
3: $\beta_i^{\text{row}} \leftarrow \beta_0(t)\big/\big(1 + b(H_i^{\text{row}})\big)$
4: $\beta_j^{\text{col}} \leftarrow \beta_0(t)\big/\big(1 + b(H_j^{\text{col}})\big)$
5: $\beta_{ij} \leftarrow \frac{1}{2}\big(\beta_i^{\text{row}} + \beta_j^{\text{col}}\big)$
6: Sample $g_{ij} \sim \text{Gumbel}(0, 1)$
7: Let $\boldsymbol{g} = (g_{ij})_{i,j}$
8: $\boldsymbol{P} \leftarrow \text{Sinkhorn}(\boldsymbol{\beta} \odot (\boldsymbol{S} + \boldsymbol{g}))$ as in Eq. (4)

---

and reduces deterministic tie-breaking effects during training. To approximate a discrete permutation, a temperature parameter $\tau \in \mathbb{R}$ is applied to the Sinkhorn operator $\boldsymbol{P} = \text{Sinkhorn}\Big(\frac{\boldsymbol{S}+\boldsymbol{g}}{\tau}\Big)$ such that as $\tau \to 0$, the resulting matrix $\boldsymbol{P}$ becomes increasingly close to a permutation matrix, while larger values of $\tau$ yield smoother, more entropic doubly stochastic matrices. During training, $\tau$ is typically annealed from a high to a low value to enable stable optimization, with smooth assignments early on and increasingly discrete assignments as training progresses.

Learning is driven by task-specific, structured losses defined directly on the reordered output. The neural network architecture is further discussed in Section 4 and Appendix E.

### 3.1. Learning permutations from structural supervision

We consider permutation learning problems where ground-truth permutations are unavailable. Instead of supervising the permutation directly, learning is driven by structural properties of the object obtained after reordering.

First, the predicted soft permutation matrix $\boldsymbol{P}_\theta \in \mathcal{B}_n$ is applied and yields a reordered object $\hat{\boldsymbol{X}} = \boldsymbol{P}_\theta \boldsymbol{X}$. The objective is a task-dependent structural loss $\mathcal{L}_{\text{struct}}(\hat{\boldsymbol{X}})$, which evaluates whether the reordered object $\hat{\boldsymbol{X}}$ satisfies desired structural properties, such as monotonicity, spatial coherence, or short tour length. The task-specific structure loss is defined in Section 4

Because structural losses typically do not uniquely identify a single permutation, multiple assignments may satisfy the loss equally well (such as a correctly assembled jigsaw puzzle rotated clockwise). As a result, different rows and columns of the soft permutation can exhibit heterogeneous uncertainty during optimization.

### 3.2. Entropy-adaptive Gumbel–Sinkhorn

**Conventions (temperature vs. inverse temperature).** We parameterize Gumbel–Sinkhorn using an inverse temperature $\beta > 0$. Throughout the paper, $\beta = \frac{1}{\tau}$ where $\tau$ denotes the conventional temperature parameter. Larger $\beta$

produces sharper (lower-entropy) soft permutations, while smaller $\beta$ yields more diffuse assignments. We write costs as $\boldsymbol{C} = -\boldsymbol{S}$. Applying inverse temperature corresponds to cost scaling: $\mathrm{Sinkhorn}(\beta \boldsymbol{S}) \equiv \mathrm{Sinkhorn}(-\beta \boldsymbol{C})$. Inverse temperature control is expressed in terms of $\beta$. When $\beta$ is a scalar, we write $\beta \boldsymbol{S}$ for standard scalar–matrix multiplication. When inverse temperature varies across entries, we use elementwise scaling and write $\boldsymbol{\beta} \odot \boldsymbol{S}$, where $\boldsymbol{\beta}, \boldsymbol{S} \in \mathbb{R}^{n \times n}$. For temperature annealing, $\beta_0(t)$ is denoted as the starting base inverse temperature, and is annealed according to a fixed schedule indexed by training iteration $t$.

A central difficulty in permutation learning is that different assignments resolve at different rates. Some rows and columns of the assignment matrix become confident early in training, while others remain highly ambiguous. A single global temperature forces all assignments to sharpen or diffuse simultaneously, either freezing incorrect matches or preventing convergence in large problems. This design is directly motivated by the theoretical limitation of global temperature control established in Section 3.3. We address this limitation by making the temperature adaptive to the uncertainty of the current soft permutation.

At each training step, we first construct a deterministic soft assignment using the base inverse temperature $\beta_0(t)$,

$$\boldsymbol{Q} = \mathrm{Sinkhorn}(\beta_0(t)\,\boldsymbol{S}).  \quad (2)$$

When constructing $\boldsymbol{Q}$ for entropy estimation, we apply a stop-gradient to the scores, i.e., $\boldsymbol{Q} = \mathrm{Sinkhorn}(\beta_0(t)\,\mathrm{stopgrad}(\boldsymbol{S}))$, so that gradients do not propagate through the entropy-based controller. Equivalently, during backpropagation within a training step, we treat the inverse-temperature field $\boldsymbol{\beta}$ as constant and differentiate only through the final mapping in Eq. (4).

From $\boldsymbol{Q}$, we compute normalized row and column entropies,

$$
\begin{aligned}
H_i^{\mathrm{row}} &= -\frac{1}{\log n} \sum_j (\boldsymbol{Q})_{ij} \, \log(\boldsymbol{Q})_{ij}, \\
H_j^{\mathrm{col}} &= -\frac{1}{\log n} \sum_i \boldsymbol{Q}_{ij} \, \log \boldsymbol{Q}_{ij}.
\end{aligned}
\quad (3)
$$

where values near zero indicate confident assignments and values near one indicate high uncertainty.

We increase temperature / decrease inverse temperature in high-entropy rows: $b(H) = b_{\max} \cdot \mathrm{clip}\left(\frac{H - H_0}{1 - H_0}, 0, 1\right)$, where $H_0 \in [0, 1)$ is an entropy threshold and $b_{\max} \geq 0$ is the maximum relative temperature increase.

These entropy estimates define an entrywise inverse-temperature field that modifies the effective assignment costs prior to Sinkhorn normalization. For each row and column, we define temperature scaling factors $\beta_i^{\mathrm{row}} =$ $\frac{\beta_0(t)}{1 + b(H_i^{\mathrm{row}})}$, $\qquad \beta_j^{\mathrm{col}} = \frac{\beta_0(t)}{1 + b(H_j^{\mathrm{col}})}$. where $b(\cdot)$ is a bounded, monotone function that increases temperature only when entropy exceeds a threshold. Row and column inverse temperatures are combined into an entrywise inverse-temperature field $\boldsymbol{\beta} \in \mathbb{R}^{n \times n}$ with entries $\beta_{ij} = \frac{1}{2}\left(\beta_i^{\mathrm{row}} + \beta_j^{\mathrm{col}}\right)$.

The final soft permutation is then sampled as

$$\boldsymbol{P} = \mathrm{Sinkhorn}(\boldsymbol{\beta} \odot (\boldsymbol{S} + \boldsymbol{g})), \, g_{ij} \sim \mathrm{Gumbel}(0, 1),  \quad (4)$$

where $\odot$ denotes elementwise multiplication.

This construction selectively reduces cost contrast in high-entropy regions by lowering inverse temperature, while leaving low-entropy regions at the base inverse temperature $\beta_0(t)$, recovering the standard Gumbel–Sinkhorn behavior in the large-inverse-temperature limit $\beta_0(t) \to \infty$. Additional implementation details are provided in Appendix N. Algorithm 1 summarizes the entropy-adaptive Gumbel–Sinkhorn method used throughout.

### 3.3. Limitations of global temperature annealing

A central assumption in existing differentiable permutation methods is that a single global temperature parameter is sufficient to control the trade-off between exploration and discretization. In practice, however, different parts of a permutation often resolve at different rates: some assignments become effectively deterministic early in training, while others remain ambiguous due to symmetry or weak structural cues. The block construction below captures this heterogeneous case in a minimal assignment problem, where the inverse temperature preferred by confident rows conflicts with the inverse temperature needed to keep ambiguous rows diffuse.

Consider the following block-ambiguous assignment problem. Let $n = n_1 + n_2$ and fix $\Delta > 0$. Define a cost matrix $\boldsymbol{C} \in \mathbb{R}^{n \times n}$ by

$$
C_{ij} = \begin{cases}
0, & i = j \leq n_1, \\
\Delta, & i \leq n_1, \, j \leq n_1, \, j \neq i, \\
0, & i = j > n_1, \\
\delta, & i > n_1, \, j > n_1, \, j \neq i, \\
M, & \text{otherwise,}
\end{cases}
\quad (5)
$$

where $M$ is any constant satisfying $M \geq 2\Delta$ (we will take $M$ large enough that cross-block mass is negligible). The top-left $n_1 \times n_1$ block is "easy" (unique diagonal minimizers separated by margin $\Delta$), while the bottom-right $n_2 \times n_2$ block is "hard" (diagonal minimizers separated by a small margin $\delta$).

**Proposition 1** (Global-temperature trade-off in a block--ambiguous assignment)**.** *Let $n = n_1 + n_2$ and let $\boldsymbol{C}$*

*be given by* (5) *with parameters* $\Delta > \delta > 0$ *and* $M$ *sufficiently large that cross-block mass is negligible. Let* $P_\beta = \text{Sinkhorn}(-\beta C)$.

*Fix any* $\varepsilon \in (0, 1/2)$ *and any* $\eta \in (0, 1/2)$. *There exist choices of* $\Delta > \delta > 0$ *(depending on* $\varepsilon, \eta, n_1, n_2$*) such that there is no* $\beta > 0$ *for which both hold:*

1. *(**easy block discretizes**) for every* $i \leq n_1$, $P_{\beta,ii} \geq 1 - \varepsilon$;

2. *(**hard block stays diffuse**) for every* $i > n_1$, $P_{\beta,ii} \leq 1 - \eta$.

Intuitively, achieving (1) requires $\beta = \Omega\big((\log(n_1/\varepsilon))/\Delta\big)$ so that the large-margin rows become nearly one-hot. In contrast, achieving (2) requires $\beta = O\big((\log((n_2 - 1)(1 - \eta)/\eta))/\delta\big)$ so that the small-margin rows do not collapse. For $\delta \ll \Delta$, these requirements are incompatible, so no single global inverse temperature can satisfy both simultaneously. A proof is provided in Appendix A.

For an illustrative comparison of global vs. entropy-adaptive inverse-temperature behavior on a block-ambiguous assignment, see Appendix M (Fig. 4).

### 3.4. Local inverse-temperature scaling of assignment costs

Entropy-adaptive temperature control can be interpreted as a form of local cost rescaling in the entropically regularized assignment. Applying Sinkhorn normalization to $\beta \odot S$ is equivalent to solving an assignment problem in which relative costs are selectively attenuated in high-uncertainty regions. This delays symmetry breaking, where assignments remain ambiguous while allowing confident regions to concentrate.

Formally, adaptive inverse temperature induces a locally rescaled cost matrix, but feasibility is still enforced globally by the doubly stochastic constraints. As a result, local temperature control cannot be reproduced by any global or separable temperature schedule. The precise optimal-transport formulation and proof of equivalence are provided in Appendix O.

**Task-specific uncertainty signals for temperature adaptation**: The entropy signal above is generic, but some tasks provide additional uncertainty measures (e.g., feasibility violations in routing). We show how to incorporate such signals purely through temperature modulation in Appendix B.1.

## 4. Experiments

We evaluate the proposed entropy-adaptive permutation learning framework on tasks where supervision is provided by the structural properties of the reordered output. Across

all experiments, the model predicts a soft permutation matrix $P$ and applies it to the input to obtain a reordered object $\hat{X} = PX$. Training losses are defined on $\hat{X}$ rather than on the permutation itself, allowing learning under weak or absent permutation supervision.

To address whether local temperature adaptation remains beneficial beyond the Birkhoff/Sinkhorn geometry, we additionally evaluated OT4P (Guo et al., 2024), an orthogonal-group relaxation with temperature-controlled matrix powering, under the same unsupervised objective. Because OT4P relies on a different relaxation geometry, its performance can be sensitive to the temperature schedule, numerical stabilizations, and the unsupervised objective. In an unsupervised sorting setting, OT4P did not recover meaningful permutations (Kendall $\tau$ near chance), as further elaborated in Appendix P.

For list sorting, the input consists of a randomly permuted list of $n$ scalar values. We follow the architecture and training protocol of (Mena et al., 2018): a one-dimensional convolutional network produces a score matrix $S \in \mathbb{R}^{n \times n}$, which is relaxed using Gumbel–Sinkhorn. The training data comprises 10,000 randomly generated lists, with 100 held-out test lists. Values are sampled uniformly from a specified range.

We consider two value regimes: $[0, 1]$, where monotonic cues are strong, and ambiguity is low, and $[10, 11]$, where relative differences between values are smaller and assignment ambiguity increases with $n$. Both supervised and unsupervised training are evaluated. Supervised training minimizes the mean squared error between the reordered list and the ground-truth sorted list. In the unsupervised setting, supervision is provided through a structural monotonicity constraint on the reordered list. For ascending order, the structural loss is defined as $\mathcal{L}_{\text{struct}}(\hat{X}) = \sum_{i=1}^{n-1} \text{ReLU}(-(\hat{x}_{i+1} - \hat{x}_i))^2$.

We compare supervised and unsupervised Gumbel–Sinkhorn with global temperature annealing, unsupervised Gumbel–Sinkhorn with entropy-adaptive temperature, and NeuralSort. NeuralSort is evaluated in a train-free manner by sweeping over temperatures and reporting the Kendall–$\tau$ score for each configuration, following common practice in differentiable sorting benchmarks. We additionally report results for a trained NeuralSort variant optimized under the same unsupervised monotonicity objective, as discussed below and in Appendix L. At evaluation time, a discrete permutation is extracted following (Mena et al., 2018) using the Hungarian algorithm, and performance is measured using Kendall–$\tau$ correlation.

Sorting results are reported in Table 1, measured by Kendall $\tau$ (higher is better). We compare supervised and unsupervised permutation learning baselines, including Gumbel–

**Table 1.** Sorting performance (Kendall $\tau$, mean $\pm$ std over 3 seeds).

| Value range | Method | 5 | 10 | 50 | 100 | 200 | 300 |
|---|---|---|---|---|---|---|---|
| | Supervised GS | $1.00 \pm 0.00$ | $1.00 \pm 0.00$ | $1.00 \pm 0.00$ | $1.00 \pm 0.00$ | $1.00 \pm 0.00$ | $1.00 \pm 0.00$ |
| | NS | $1.00 \pm 0.00$ | $1.00 \pm 0.00$ | $1.00 \pm 0.00$ | $\mathbf{1.00 \pm 0.00}$ | $\mathbf{0.99 \pm 0.00}$ | $\mathbf{0.73 \pm 0.00}$ |
| $[0, 1]$ | Trained NS | $0.94 \pm 0.10$ | $0.71 \pm 0.50$ | $0.17 \pm 0.80$ | $0.00 \pm 0.87$ | $-0.05 \pm 0.91$ | $-0.03 \pm 0.89$ |
| | Unsupervised GS (Ours) | $1.00 \pm 0.00$ | $1.00 \pm 0.00$ | $1.00 \pm 0.00$ | $1.00 \pm 0.00$ | $0.87 \pm 0.22$ | $0.44 \pm 0.22$ |
| | Unsupervised GS w TA (Ours) | $1.00 \pm 0.00$ | $1.00 \pm 0.00$ | $1.00 \pm 0.00$ | $1.00 \pm 0.00$ | $0.91 \pm 0.16$ | $0.69 \pm 0.23$ |
| | Supervised GS | $1.00 \pm 0.00$ | $1.00 \pm 0.00$ | $1.00 \pm 0.00$ | $1.00 \pm 0.00$ | $1.00 \pm 0.00$ | $1.00 \pm 0.00$ |
| | NS | $1.00 \pm 0.00$ | $0.33 \pm 0.00$ | $0.01 \pm 0.00$ | $-0.00 \pm 0.00$ | $-0.00 \pm 0.00$ | $0.00 \pm 0.00$ |
| $[10, 11]$ | Trained NS | $0.21 \pm 0.68$ | $0.03 \pm 0.85$ | $-0.38 \pm 0.13$ | $-0.10 \pm 0.42$ | $-0.07 \pm 0.31$ | $-0.04 \pm 0.28$ |
| | Unsupervised GS (Ours) | $1.00 \pm 0.00$ | $1.00 \pm 0.00$ | $1.00 \pm 0.00$ | $0.74 \pm 0.10$ | $0.60 \pm 0.22$ | $0.06 \pm 0.12$ |
| | Unsupervised GS w TA (Ours) | $1.00 \pm 0.00$ | $1.00 \pm 0.00$ | $1.00 \pm 0.00$ | $\mathbf{0.80 \pm 0.11}$ | $\mathbf{0.67 \pm 0.22}$ | $\mathbf{0.15 \pm 0.14}$ |

**Table 2.** Jigsaw results on CelebA-Test (unsupervised training). We report Kendall $\tau$ (mean $\pm$ std over seeds) and improvement over a mask-only baseline, $\Delta\tau = \tau_{\mathrm{method}} - \tau_{\mathrm{mask\text{-}only}}$. Mask-only baselines (anchor + random fill) are 0.029 ($5 \times 5$), 0.126 ($6 \times 6$), and 0.189 ($7 \times 7$). See Appendix D.

| | $5 \times 5$ (25, 1 anchor) | | $6 \times 6$ (36, 6 anchors) | | $7 \times 7$ (49, 12 anchors) | |
|---|---|---|---|---|---|---|
| Method | Kendall $\tau$ | $\Delta\tau \uparrow$ | Kendall $\tau$ | $\Delta\tau \uparrow$ | Kendall $\tau$ | $\Delta\tau \uparrow$ |
| GS (global anneal) | $0.178 \pm 0.003$ | $0.149 \pm 0.003$ | $0.256 \pm 0.003$ | $0.130 \pm 0.003$ | $0.201 \pm 0.003$ | $0.012 \pm 0.003$ |
| GS (entropy-adaptive) | $\mathbf{0.199 \pm 0.010}$ | $\mathbf{0.170 \pm 0.010}$ | $\mathbf{0.283 \pm 0.038}$ | $\mathbf{0.157 \pm 0.038}$ | $\mathbf{0.291 \pm 0.093}$ | $\mathbf{0.102 \pm 0.093}$ |

Sinkhorn (GS) and NeuralSort (NS), and TA denotes entropy-adaptive temperature control (ours). NS is evaluated only in the unsupervised setting, either without training or trained under the same structural objective as GS. For GS, we compare a global temperature schedule to the proposed entropy-adaptive temperature control.

In the low-ambiguity regime $[0, 1]$, all methods perform well up to moderate problem sizes, and differences between temperature control strategies are small. In contrast, under the high-ambiguity regime $[10, 11]$, performance degrades as $n$ increases. In this setting, entropy-adaptive temperature consistently outperforms global annealing, with the gap widening as $n$ increases, indicating improved robustness when uncertainty is heterogeneous. In an additional hard sorting schedule study at $n = 300$ with values in $[10, 11]$, entropy-adaptive GS improved Kendall $\tau$ over global GS by $+0.122$, $+0.100$, and $+0.059$ under annealing horizons of 150, 100, and 50 epochs, respectively, indicating that the gains are not explained only by choosing a better scalar schedule. We further observe that operator-based sorting relaxations degrade rapidly under this weak unsupervised signal, consistent with their reliance on a scalar ordering prior rather than a latent permutation representation (Appendix L). This effect is visualized in Figure 2, which plots Kendall $\tau$ against assignment ambiguity during training and shows that entropy-adaptive temperature achieves strictly better performance at comparable entropy levels.

We next evaluate image reordering, in which an image is partitioned into tiles and randomly permuted, and the model predicts a permutation that reconstructs the original spatial arrangement. Supervised training minimizes reconstruction error with respect to the ground-truth ordering, while unsupervised training relies on a smoothness prior that penalizes visual discontinuities across adjacent tile borders. Formally, $\mathcal{L}_{\mathrm{struct}}(\hat{\boldsymbol{X}}) = \sum_{\mathrm{adjacent\ tiles}} \|\partial\hat{\boldsymbol{X}}\|^2$. Here $\partial\hat{\boldsymbol{X}}$ denotes the image gradient across shared boundaries of adjacent tiles, and $\|\cdot\|^2$ is the squared $\ell_2$ norm over pixel differences along each boundary.

We compare constant temperature, global annealing, and entropy-adaptive temperature control. Performance is measured using Kendall–$\tau$ correlation between predicted and ground-truth tile positions, and the test score. Results are reported in Tables 4 and 2. On CelebA-Test with $6 \times 6$ puzzles under unsupervised training, global temperature annealing improves over fixed-temperature baselines, but entropy-adaptive temperature control achieves the highest Kendall $\tau$ and more stable convergence across seeds (Table 4). Across puzzle sizes, the advantage of entropy-adaptive temperature grows with problem size (Table 2), indicating increased robustness as assignment ambiguity becomes more heterogeneous.

**Table 3.** Tour length and relative optimality gap (%) for unsupervised TSP (mean $\pm$ std over 3 seeds).

| Model | Method | $n = 100$ | | $n = 200$ | | $n = 500$ | | $n = 1000$ | |
|---|---|---|---|---|---|---|---|---|---|
| | | Length $\downarrow$ | Gap (%) $\downarrow$ | Length $\downarrow$ | Gap (%) $\downarrow$ | Length $\downarrow$ | Gap (%) $\downarrow$ | Length $\downarrow$ | Gap (%) $\downarrow$ |
| | UTSP (Min et al., 2023) | $22.53 \pm 0.54$ | $45.08 \pm 3.48$ | $32.33 \pm 0.56$ | $50.68 \pm 2.61$ | $55.26 \pm 3.05$ | $66.61 \pm 9.19$ | $82.43 \pm 2.69$ | $77.44 \pm 5.79$ |
| GNN | GS UTSP (Min & Gomes, 2023; 2025) | $20.38 \pm 0.60$ | $31.24 \pm 3.86$ | $29.72 \pm 0.21$ | $38.51 \pm 0.98$ | $50.73 \pm 1.72$ | $52.95 \pm 5.18$ | $79.41 \pm 1.39$ | $70.95 \pm 3.03$ |
| | GS UTSP + DV-TA (Ours) | $\mathbf{20.21 \pm 0.23}$ | $\mathbf{30.14 \pm 1.48}$ | $\mathbf{29.54 \pm 0.25}$ | $\mathbf{37.67 \pm 1.17}$ | $\mathbf{49.44 \pm 0.15}$ | $\mathbf{49.06 \pm 0.45}$ | $\mathbf{77.88 \pm 1.39}$ | $\mathbf{67.65 \pm 3.00}$ |

Because anchor tiles artificially inflate Kendall $\tau$, we additionally interpret results relative to a mask-only baseline that uses anchor information alone (Appendix D). Measured relative to this baseline, entropy-adaptive temperature yields larger gains than global annealing, particularly for larger puzzles, confirming that improvements arise from better optimization of unconstrained tiles rather than from anchor supervision.

For larger puzzles, the unsupervised boundary-consistency objective is highly ambiguous due to repeated textures and local symmetries, and we observe non-convergence across all methods when no anchor information is provided. We therefore fix a small number of anchor tiles in their correct positions to break symmetries and make the optimization well-posed, while remaining predominantly unsupervised ($1/25$ for $5 \times 5$, $6/36$ for $6 \times 6$, and $12/49$ for $7 \times 7$). The anchor budget is chosen as the smallest value that avoids degenerate failure modes across seeds. Because anchors fix many pairwise relations, raw Kendall $\tau$ is inflated by the amount of revealed information and is not directly comparable across puzzle sizes. To account for this effect, we additionally report improvements over a mask-only baseline that correctly places the anchor tiles and uniformly randomizes the remaining positions (Appendix D).

Finally, we apply the framework to unsupervised TSP following the UTSP experimental protocol (Min et al., 2023). In this unsupervised setting, the structural objective is defined on the induced tour. $\mathcal{L}_{\text{struct}}(\hat{\boldsymbol{X}}) = \sum_{i=1}^{n} \|\hat{x}_i - \hat{x}_{i+1}\|_2, \hat{x}_{n+1} = \hat{x}_1$. A graph neural network predicts a permutation that orders cities into a tour. Unlike prior work, we do not apply a post-hoc solver; we report the tour length induced directly by the learned permutation. We evaluate a degree-violation-based temperature adaptation (DV-TA) that increases temperature for nodes whose expected degree deviates from two. The same qualitative ordering persists at $n = 1000$: GS improves over the UTSP baseline, and adding DV-TA further reduces both tour length and optimality gap without additional numerical instability in this regime.

**Sensitivity to temperature adaptation hyperparameters.** We evaluate the sensitivity of entropy-adaptive temperature control to the entropy threshold $H_0$ and maximum boost $b_{\max}$. Results show that performance is stable across a wide range of values, with no brittle behavior or training failures. We report a detailed sensitivity analysis for unsupervised TSP in Appendix K.

Evaluation is performed using the average tour length and the relative optimality gap with respect to the Concorde-optimal solutions. Results are reported in Table 3. DV-TA provides consistent gains, particularly for larger problem sizes. Additional UTSP ablations are reported in Appendix C.

**Table 4.** Unsupervised Jigsaw ablation on CelebA-Test with $7 \times 7$ tiles ($n = 49$, 12 anchors). Variants differ in the smoothness loss or in how row/column entropy is combined for TA; see Appendix Q.1.

| Method | Kendall $\tau \uparrow$ |
|---|---|
| GS (fixed $\beta = 2$) | $0.152 \pm 0.025$ |
| GS (fixed $\beta = 1$) | $0.194 \pm 0.032$ |
| GS (global anneal) | $0.197 \pm 0.005$ |
| GS (entropy-adaptive) *product* | $0.246 \pm 0.033$ |
| GS (entropy-adaptive) *smooth2k* | $0.247 \pm 0.051$ |
| GS (entropy-adaptive) | $\mathbf{0.291 \pm 0.093}$ |

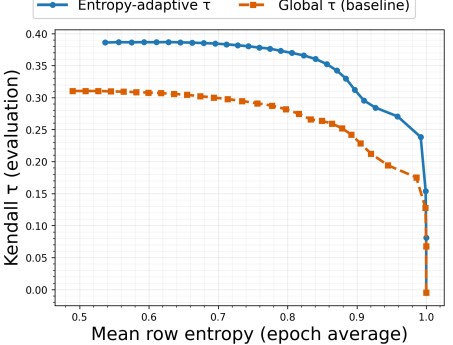

**Figure 2.** Sorting performance (Kendall $\tau$) as a function of assignment ambiguity, measured by the mean row entropy of the soft permutation, with curves tracing optimization progress over training. For a given level of ambiguity, entropy-adaptive temperature consistently attains higher Kendall $\tau$ than a global temperature baseline, indicating a strictly better ambiguity–performance trade-off.

**Implementation details.** We defer all optimization, normalization, and hyperparameter details to Appendix E.

## 5. Conclusion

We studied permutation learning in settings where supervision is available only through structure in the reordered output, and identified heterogeneous uncertainty across assignments as a central obstacle to scalability in differentiable permutation methods. Standard Gumbel–Sinkhorn relies on a single global temperature, forcing all rows and columns to sharpen simultaneously and inducing an unavoidable trade-off between stability and discretization as problem size and ambiguity increase. To address this limitation, we introduced an entropy-adaptive variant of Gumbel–Sinkhorn that replaces the global temperature with a row- and column-dependent inverse-temperature field derived from assignment entropy. This allows confident assignments to discretize early while preserving exploration in ambiguous regions. We further provided an optimal-transport interpretation in which adaptive temperature corresponds to local rescaling of assignment costs under global doubly stochastic constraints. Across sorting, unsupervised jigsaw reconstruction, and unsupervised TSP-style routing, entropy-adaptive temperature control consistently improved convergence reliability and final permutation quality relative to fixed or globally annealed baselines, with the largest gains appearing

in large-scale and highly ambiguous regimes. These results suggest that effective permutation learning from structure depends critically on managing where and when entropy collapses during optimization.

**Limitations:** Our approach introduces additional hyperparameters for entropy adaptation, such as the entropy threshold and maximum temperature adjustment, which require tuning, although we find performance to be stable across a wide range of values. Moreover, while local entropy control improves optimization stability, it does not resolve fundamental ambiguities in objectives that are highly symmetric or weakly informative, and some form of symmetry breaking or additional structure would still be required in such settings.

**Future work** could investigate learning entropy-adaptive temperature control end-to-end, rather than relying on fixed controller forms or thresholds. Another direction is to extend uncertainty-aware discretization to more complex combinatorial structures, such as partial permutations or constrained routing problems, and to analyze its theoretical properties beyond the assignment setting.

**Acknowledgments.** OL and RE were supported by the MOST grant No. 0007341.

# Impact Statement

Learning permutations from structure has potential benefits in domains where correct orderings are latent, ambiguous, or costly to annotate, including document and event sequencing, visual matching and alignment, and routing or scheduling problems subject to global constraints. By improving the stability and scalability of differentiable permutation relaxations, entropy-adaptive temperature control may reduce reliance on labeled orderings, lower computational waste from unstable training runs, and make permutation-based components more practical in large systems. At the same time, permutation learning can be applied in sensitive settings, such as ranking, matching, or prioritization, where learned orderings may reflect biases encoded in the data or in the structural objectives. Uncertainty-aware discretization may also cause early spurious structure to become fixed if objectives are misspecified or if symmetry-breaking signals are misleading. For these reasons, downstream applications in consequential domains should include careful objective design, bias and robustness evaluations, and analysis of failure modes under distribution shift. When discrete permutations are decoded for decision-making, uncertainty and calibration diagnostics should be reported alongside point estimates to support responsible use.

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

# A. Proof of Proposition 1

We prove Proposition 1 for the block-ambiguous cost matrix defined in (5). Let $n = n_1 + n_2$ and recall that

$$C_{ij} = \begin{cases} 0, & i = j \leq n_1, \\ \Delta, & i \leq n_1, \ j \leq n_1, \ j \neq i, \\ 0, & i = j > n_1, \\ \delta, & i > n_1, \ j > n_1, \ j \neq i, \\ M, & \text{otherwise.} \end{cases}$$

We write $\boldsymbol{P}_\beta = \text{Sinkhorn}(-\beta \boldsymbol{C})$ for the (unique, for $\varepsilon_{\text{ent}} > 0$) minimizer of

$$\min_{\boldsymbol{P} \in \mathcal{B}_n} \ \langle \boldsymbol{C}, \boldsymbol{P} \rangle + \varepsilon_{\text{ent}} \sum_{i,j} P_{ij} \log P_{ij}, \qquad \varepsilon_{\text{ent}} = \tfrac{1}{\beta}.$$

**Step 1: Discretizing the easy block requires a large $\beta$.**

**Lemma 1** (Easy-block discretization). *If $P_{\beta,ii} \geq 1 - \epsilon$ for all $i \leq n_1$, then necessarily*

$$\beta \ \geq \ \frac{1}{\Delta} \log \frac{n_1 - 1}{\epsilon}.$$

*Proof.* Fix $i \leq n_1$. Within row $i$, the diagonal entry has cost 0, while all other entries $j \leq n_1$, $j \neq i$, have cost $\Delta$. Writing the Sinkhorn solution as $P_{\beta,ij} = u_i K_{ij} v_j$ with $K_{ij} = e^{-\beta C_{ij}}$, we have $K_{ii} = 1$ and $K_{ij} \leq e^{-\beta \Delta}$ for $j \neq i$. Thus

$$\sum_{j \neq i, \, j \leq n_1} P_{\beta,ij} \leq u_i e^{-\beta \Delta} \sum_{j \leq n_1} v_j.$$

Since $\sum_{j \leq n_1} P_{\beta,ij} \leq 1$ and $P_{\beta,ii} = u_i v_i$, a sufficient condition for $P_{\beta,ii} \geq 1 - \epsilon$ is

$$(n_1 - 1) e^{-\beta \Delta} \leq \epsilon,$$

which yields the stated bound on $\beta$. $\qquad\square$

**Step 2: Keeping the hard block diffuse requires a small $\beta$.**

**Lemma 2** (Hard-block non-discretization). *Assume $\boldsymbol{P}_\beta$ assigns zero mass to cross-block entries (equivalently, restrict the problem to the bottom-right $n_2 \times n_2$ block). If $P_{\beta,ii} \leq 1 - \eta$ for all $i > n_1$, then necessarily*

$$\beta \ \leq \ \frac{1}{\delta} \log \frac{(n_2 - 1)(1 - \eta)}{\eta}.$$

*Proof.* Restrict attention to the bottom-right $n_2 \times n_2$ block. By symmetry of the costs within this block (all diagonals equal, all off-diagonals equal), the Sinkhorn solution on this block has constant diagonal entries $a$ and constant off-diagonal entries $b$ with

$$a + (n_2 - 1)b = 1, \qquad \frac{a}{b} = e^{\beta \delta},$$

because the Gibbs kernel satisfies $K_{ii} = 1$ and $K_{ij} = e^{-\beta \delta}$ for $i \neq j$, and Sinkhorn scaling preserves equality classes under the permutation symmetries of the block. Solving these two equations gives

$$a \ = \ \frac{1}{1 + (n_2 - 1)e^{-\beta \delta}}.$$

The condition $a \leq 1 - \eta$ is equivalent to

$$\frac{1}{1 + (n_2 - 1)e^{-\beta \delta}} \leq 1 - \eta \quad \Longleftrightarrow \quad (n_2 - 1)e^{-\beta \delta} \geq \frac{\eta}{1 - \eta},$$

which yields

$$\beta \ \leq \ \frac{1}{\delta} \log \frac{(n_2 - 1)(1 - \eta)}{\eta}.$$

$\qquad\square$

**Step 3: Incompatibility of requirements.** Suppose for contradiction that there exists $\beta > 0$ such that:

1. $P_{\beta,ii} \geq 1 - \epsilon$ for all $i \leq n_1$;

2. $P_{\beta,ii} \leq 1 - \eta$ for all $i > n_1$.

By Lemma 1, condition (1) implies

$$\beta \;\geq\; \frac{1}{\Delta} \log \frac{n_1 - 1}{\epsilon}.$$

By Lemma 2, condition (2) implies

$$\beta \;\leq\; \frac{1}{\delta} \log \frac{(n_2 - 1)(1 - \eta)}{\eta}.$$

Choose $\Delta > \delta > 0$ so that

$$\frac{1}{\Delta} \log \frac{n_1 - 1}{\epsilon} \;>\; \frac{1}{\delta} \log \frac{(n_2 - 1)(1 - \eta)}{\eta}.$$

Then no $\beta$ can satisfy both inequalities, a contradiction. Hence, no such $\beta$ exists. $\qquad\square$

## B. Proof of Proposition 2

*Proof.* Let $\boldsymbol{K} = \exp(\boldsymbol{\beta} \odot \boldsymbol{S})$, so that $K_{ij} = \exp(-\beta_{ij} C_{ij})$ where $\boldsymbol{C} = -\boldsymbol{S}$. Sinkhorn normalization computes the unique doubly stochastic matrix

$$\boldsymbol{P} = \arg \min_{\boldsymbol{P} \in \mathcal{B}_n} \mathrm{KL}(\boldsymbol{P} \,\|\, \boldsymbol{K}),$$

where KL denotes the Kullback–Leibler divergence. Expanding the KL objective and discarding constants yields

$$\mathrm{KL}(\boldsymbol{P} \,\|\, \boldsymbol{K}) = \sum_{i,j} P_{ij} \log P_{ij} + \langle \boldsymbol{\beta} \odot \boldsymbol{C}, \, \boldsymbol{P} \rangle.$$

Since the negative entropy term is strictly convex on the interior of $\mathcal{B}_n$, the objective admits a unique minimizer, which coincides with the Sinkhorn output. $\qquad\square$

### B.1. Task-specific uncertainty signals for temperature adaptation

The adaptive temperature mechanism described above relies on the entropy of the soft permutation as a generic proxy for uncertainty. While this signal is sufficient for many permutation learning problems, some structured tasks admit additional notions of uncertainty or infeasibility that are not fully captured by assignment entropy alone. Importantly, such signals do not alter the permutation formulation or the learning objective; instead, they provide complementary information about where additional exploration is needed. We show that these task-specific signals can be incorporated into the adaptive entropy framework solely through temperature modulation.

More generally, let $\boldsymbol{Q} \in \mathbb{R}^{n \times n}$ denote the deterministic soft permutation obtained from Sinkhorn normalization at base temperature $\beta_0(t)$. We assume access to a task-dependent uncertainty signal $u_i \geq 0$, defined over input elements or positions, which measures structural inconsistency or constraint violation under the current soft assignment. These signals are normalized and converted into temperature scaling factors,

$$s_i = 1 + b_{\max} \, \mathrm{norm}(u_i), \tag{6}$$

where $\mathrm{norm}(\cdot)$ rescales values to $[0, 1]$ and $b_{\max}$ bounds the maximum temperature increase. Row- and column-level scalings are then combined

$$\beta_{ij} = \beta_0(t) \big/ \phi(s_i^{\mathrm{row}}, s_j^{\mathrm{col}}).$$

All other aspects of the permutation learning procedure remain unchanged.

**Degree-based uncertainty in routing problems.** In routing problems such as the Traveling Salesman Problem (TSP), each node must have a degree of exactly two in the induced tour. Violations of this constraint are a dominant source of failure and

may persist even when permutation entropies are low. Let $E \in [0,1]^{n \times n}$ denote a soft edge adjacency matrix induced by the current permutation, and let

$$d_i = \sum_{j \neq i} E_{ij},$$

denote the expected degree of node $i$. We define a degree-violation signal $u_i = |d_i - 2|$, which increases temperature for structurally problematic nodes and encourages additional exploration in the corresponding rows and columns.

## C. Additional UTSP ablations

Table 5 reports an additional UTSP ablation comparing global annealing (TA) to the proposed degree-violation-based adaptation (DV-TA). We observe that GS UTSP + TA provides only marginal improvement over the GS UTSP baseline across problem sizes, whereas DV-TA yields consistent gains. This supports our choice to focus the main text on DV-TA as the task-specific temperature adaptation for routing problems.

| Method | $n = 100$ | | $n = 200$ | | $n = 500$ | |
|---|---|---|---|---|---|---|
| | Length | Gap (%) | Length | Gap (%) | Length | Gap (%) |
| GS UTSP (Min & Gomes, 2023; 2025) | $20.38 \pm 0.60$ | $31.24 \pm 3.86$ | $29.72 \pm 0.21$ | $38.51 \pm 0.98$ | $50.73 \pm 1.72$ | $52.95 \pm 5.18$ |
| GS UTSP + TA (global) (Ours) | $20.36 \pm 0.53$ | $31.11 \pm 3.41$ | $29.66 \pm 0.23$ | $38.23 \pm 1.07$ | $50.54 \pm 1.36$ | $52.38 \pm 4.10$ |
| GS UTSP + DV-TA (Ours) | $\mathbf{20.21 \pm 0.23}$ | $\mathbf{30.14 \pm 1.48}$ | $\mathbf{29.54 \pm 0.25}$ | $\mathbf{37.67 \pm 1.17}$ | $\mathbf{49.44 \pm 0.15}$ | $\mathbf{49.06 \pm 0.45}$ |

**Table 5.** UTSP ablation: global temperature annealing (TA) yields marginal gains over GS UTSP, while DV-TA provides consistent improvements (mean $\pm$ std over 3 seeds).

## D. Mask-only baseline for anchored jigsaw puzzles

As discussed in Section 4, a small number of anchor tiles is fixed at their correct positions to break symmetries and make large unsupervised jigsaw puzzles well-posed. While anchors are necessary for optimization to succeed, they also inflate Kendall $\tau$ by fixing many pairwise relations independently of learning. To quantify this effect, we report a *mask-only* baseline that measures the expected Kendall $\tau$ achievable using the anchor information alone.

Consider a jigsaw puzzle with $n$ tiles and an anchor set of size $k$. Let $\pi^\star$ denote the ground-truth permutation. The mask-only baseline is constructed as follows:

1. Select an anchor set $F \subseteq \{1, \ldots, n\}$ with $|F| = k$. Unless otherwise stated, anchor positions are sampled uniformly at random.

2. Place all anchor tiles at their correct positions according to $\pi^\star$.

3. Assign the remaining $n - k$ tiles to the remaining positions using a uniform random permutation.

4. Compute Kendall $\tau$ between the resulting permutation and $\pi^\star$.

This procedure uses exactly the same anchor information available to the learning methods but does not exploit any structural cues from image content or boundary consistency.

The reported values are Monte Carlo estimates of the expected Kendall $\tau$ under the above process. For each puzzle size and anchor budget, we repeat the mask-only construction over many independent trials and report the empirical mean. Formally,

$$\mathbb{E}[\tau] \approx \frac{1}{R} \sum_{r=1}^{R} \tau^{(r)},$$

where $\tau^{(r)}$ is the Kendall $\tau$ obtained in trial $r$ and $R$ is the number of Monte Carlo samples.

Table 6 reports the estimated mask-only baseline Kendall $\tau$ for the anchor regimes used in the main experiments. As expected, the baseline increases with the number of anchors, even though the unanchored tiles are placed randomly. This explains why raw Kendall $\tau$ values are not directly comparable across puzzle sizes with different anchor budgets.

The mask-only baseline isolates the portion of Kendall $\tau$ attributable solely to revealing anchor tiles. Method performance can therefore be interpreted relative to this baseline, for example by reporting an improvement $\Delta \tau = \tau_{\text{method}} - \tau_{\text{mask-only}}$.

| Baseline | $5 \times 5$ (25, 1 anchor) | $6 \times 6$ (36, 6 anchors) | $7 \times 7$ (49, 12 anchors) |
|---|---|---|---|
| mask-only (anchor + random fill) | 0.029 | 0.126 | 0.189 |

**Table 6.** Mask-only baseline Kendall $\tau$ obtained by placing anchor tiles correctly and filling remaining positions uniformly at random. Values are Monte Carlo estimates under randomly sampled anchor positions. The exact baseline depends on the anchor placement scheme, but the monotonic increase with anchor budget is consistent across choices.

This correction removes the artificial advantage conferred by larger anchor budgets and clarifies that the gains observed in the main text arise from improved optimization of the remaining unconstrained tiles, rather than from additional supervision.

## E. Detailed Experimental Setup

**Implementation details.** All experiments use log-space Sinkhorn normalization with a fixed number of iterations (10 for sorting and jigsaw, 50 for TSP) and enforce doubly stochastic constraints. Training uses Gumbel–Sinkhorn sampling with multiple samples averaged in the loss, while evaluation is deterministic using Hungarian decoding. Global-temperature baselines anneal a single temperature over training, whereas entropy-adaptive methods compute normalized row and column entropies from the soft permutation $Q$ and apply bounded row- and column-wise temperature scaling after a short warm-up. All models are trained with Adam, gradient clipping at norm 1.0. We report the averaged kendal-$\tau$ over three random seeds.

## F. Experimental Protocol

In all experiments, discrete permutations are extracted using Hungarian decoding. All reported results are averaged over three random seeds unless otherwise stated.

Random seeds affect model initialization, data shuffling, and stochastic sampling. Sinkhorn normalization is performed in log space with a fixed number of iterations and no convergence tolerance.

## G. Unsupervised Traveling Salesman Problem (TSP)

### G.1. Data and Evaluation

City coordinates are loaded from pre-generated instances and normalized by subtracting the mean and applying a fixed rescaling factor. Tour length is computed as the sum of Euclidean distances between consecutive cities, including the return to the start. Kendall–$\tau$ is computed using inversion count with canonicalization over forward and reverse tours.

### G.2. Model and Loss

The GS-based model predicts a soft permutation over cities, which directly induces a tour without post-hoc search. Training minimizes the expected tour length computed from the induced successor heatmap. A row-sum constraint penalty is included in the loss. Degree-violation signals are used exclusively for temperature adaptation and do not directly affect the loss.

### G.3. Optimization and Relaxation

Log-space Sinkhorn normalization is applied with 50 iterations. Training uses Gumbel–Sinkhorn sampling with 8 samples per batch for GS-based models and 1 sample for UTSP. Evaluation is deterministic and does not use Gumbel noise. Optimization uses Adam with gradient clipping at norm 1.0 and no weight decay.

### G.4. Hyperparameters

| $n$ | Batch | Epochs | $\beta_0$ schedule | Adapt start | $H_0/b_{\max}$ |
|---|---|---|---|---|---|
| 100 | 128 | 300 | $1.0 \rightarrow 5$ (200 ep, linear) | 30 | 0.65/0.10 |
| 200 | 128 | 300 | $1.0 \rightarrow 5$ (200 ep, linear) | 30 | 0.65/0.10 |
| 500 | 8 | 500 | $0.285 \rightarrow 2$ (300 ep, linear) | 30 | 0.65/0.10 |
| 1000 | 8 | 500 | $0.285 \rightarrow 2$ (300 ep, linear) | 30 | 0.65/0.10 |

**Table 7.** Hyperparameters for unsupervised TSP experiments.

# H. Number Sorting

### H.1. Data and Evaluation

Training data consists of randomly generated lists of scalar values sampled uniformly from either $[0, 1]$ or $[10, 11]$. Each run uses 10,000 training examples and 100 test examples.

### H.2. Model and Loss

The model is a lightweight Conv1D network that outputs assignment scores. Supervised training minimizes mean squared error between reordered outputs and ground-truth sorted lists. Unsupervised training minimizes a monotonicity violation loss computed from adjacent differences in the reordered sequence.

### H.3. Optimization and Relaxation

Log-space Sinkhorn normalization is applied with 10 iterations. Training uses Gumbel–Sinkhorn sampling with 5 samples per batch. Evaluation uses deterministic Hungarian decoding. Models are trained with Adam and no learning-rate scheduling.

### H.4. Hyperparameters

| $n$ | Batch | Epochs | $\beta_0$ schedule | Adapt start | $H_0/b_{\max}$ |
|---|---|---|---|---|---|
| 5–10 | 256 | 150 | $0.66 \to 2$ (linear) | 1 | 0.7/0.1 |
| 50–100 | 256 | 150 | $0.66 \to 2$ (linear) | 1 | 0.7/0.1 |
| 200–300 | 256 | 150 | $0.66 \to 2$ (linear) | 1 | 0.7/0.1 |

**Table 8.** Hyperparameters for number sorting experiments.

# I. Jigsaw Reconstruction

### I.1. Data and Evaluation

Experiments are conducted on CelebA images resized to $178 \times 178$. Images are split into $p \times p$ grids with $p \in \{4, 5, 6, 7\}$.

### I.2. Model and Loss

A convolutional network predicts assignment scores for each tile. Unsupervised training minimizes a boundary smoothness loss between adjacent tiles. To break symmetries in larger puzzles, a small number of anchor tiles are fixed at known positions.

### I.3. Optimization and Relaxation

Log-space Sinkhorn normalization is applied with 10 iterations. Training uses 5 Gumbel–Sinkhorn samples per batch. Evaluation uses deterministic Hungarian decoding.

### I.4. Hyperparameters

| $p$ | $n$ | Anchors | Epochs | $\beta_0$ schedule | Adapt start | $H_0/b_{\max}$ |
|---|---|---|---|---|---|---|
| 4 | 16 | 0 | 500 | $0.5 \to 2$ (linear) | 1 | 0.6/0.35 |
| 5 | 25 | 1 | 500 | $0.5 \to 2$ (linear) | 1 | 0.7/0.20 |
| 6 | 36 | 6 | 600 | $0.5 \to 1.33$ (300 ep, exp) | 1 | 0.85/0.35 |
| 7 | 49 | 12 | 600 | $0.5 \to 1.66$ (600 ep, exp) | 1 | 0.85/0.35 |

**Table 9.** Hyperparameters for unsupervised jigsaw experiments.

# J. Runtime Overhead

Entropy-adaptive temperature control requires one additional deterministic Sinkhorn normalization per training step to estimate assignment entropy. This increases training time by approximately 10–15% relative to global-temperature baselines, with negligible additional memory overhead.

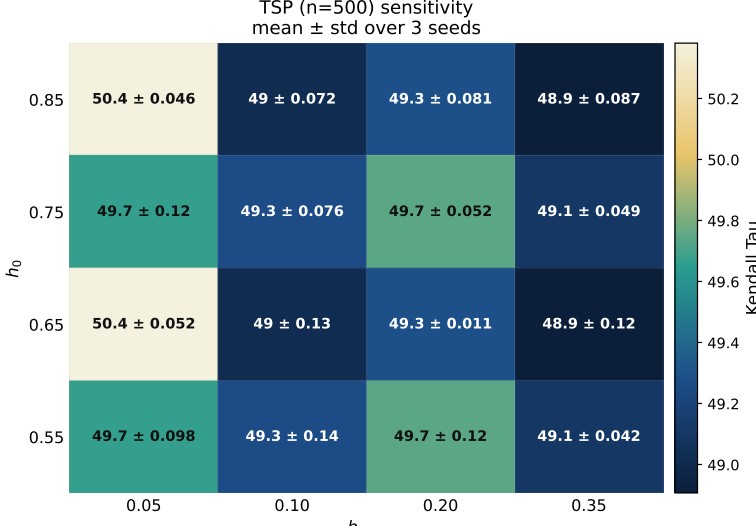

**Figure 3.** Sensitivity of entropy-adaptive temperature control to the entropy threshold $H_0$ and maximum boost $b_{\max}$ on unsupervised TSP ($n = 500$). Values indicate the test evaluation tour length; all configurations converge without failures.

## K. Sensitivity Analysis of Entropy-Adaptive Temperature Control

We analyze the sensitivity of entropy-adaptive temperature control to its two main hyperparameters: the entropy threshold $H_0$ and the maximum relative temperature increase $b_{\max}$. The threshold $H_0$ determines when local temperature boosting activates, while $b_{\max}$ controls how much additional exploration is permitted in high-uncertainty regions.

We perform a grid sweep over $H_0 \in \{0.55, 0.65, 0.75, 0.85\}$ and $b_{\max} \in \{0.05, 0.10, 0.20, 0.35\}$, holding all other components fixed. Experiments are conducted on unsupervised TSP with $n = 500$ nodes using the GS-based UTSP model with entropy-adaptive temperature control. For each configuration, we report the test tour length. A run is considered failed if it produces NaNs or does not improve over the initialization; no such failures were observed.

Figure 3 shows that performance varies smoothly across the $(H_0, b_{\max})$ grid, with no sharp degradation or unstable regions. All tested configurations yield comparable tour lengths, indicating that the adaptive mechanism does not rely on precise tuning of these hyperparameters. This robustness is expected in routing problems, where feasibility constraints strongly restrict the space of valid permutations once local exploration is preserved.

We observe mild trends consistent with the interpretation of the hyperparameters: very small $b_{\max}$ values can under-preserve exploration in ambiguous regions, while overly large $b_{\max}$ may slightly delay convergence. Similarly, lower $H_0$ activates temperature boosting more broadly, whereas higher $H_0$ restricts adaptation to only highly uncertain rows and columns. However, these effects are secondary compared to enabling local, uncertainty-aware temperature control itself.

Across all tested settings, we find that choosing $H_0$ in the range $[0.65, 0.85]$ and $b_{\max}$ in $[0.1, 0.35]$ yields stable and competitive performance. The default values used in the main experiments are selected from this flat region of the sensitivity landscape.

## L. On Differentiable Sorting Operators as Unsupervised Baselines

Differentiable sorting operators such as NeuralSort and SoftSort provide continuous relaxations of the `argsort` operation by construction. These methods assume that the task can be solved by predicting a scalar key for each element and applying a (soft) sorting operator to those keys. As a result, the permutation is not learned as a latent object, but is instead fully determined by the induced scalar ordering.

This inductive bias is well suited to supervised or strongly guided settings, where the correct ordering is directly specified or can be inferred reliably from labels. In contrast, our setting treats the permutation as a latent operator and relies only on weak structural supervision defined on the reordered output (e.g., monotonicity violations). Under such objectives, the scalar-ordering assumption fundamentally alters the learning dynamics. If a simple projection suffices to approximately satisfy the structural constraint, the sorting operator enforces an ordering early and suppresses further learning. Conversely,

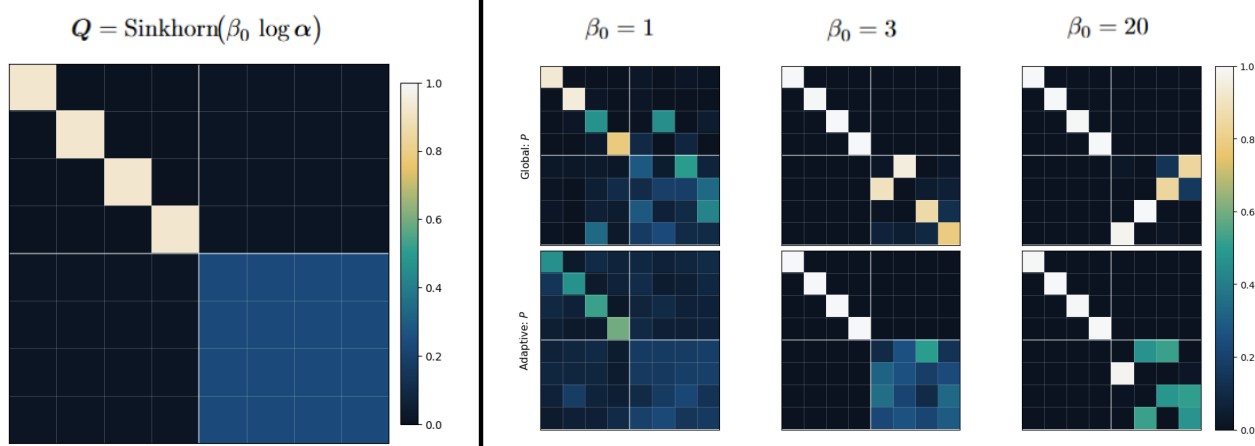

**Figure 4.** Illustration of global versus entropy-adaptive Gumbel–Sinkhorn on a block-ambiguous assignment problem. **Left:** Deterministic soft permutation $Q = \text{Sinkhorn}(\beta_0 \, S)$ with base inverse temperature $\beta_0 = 1$. **Right:** Samples under increasing base inverse temperature $\beta_0 \in \{1, \, 3, \, 20\}$. Global inverse temperature sharpens all entries uniformly; entropy-adaptive control preserves exploration in ambiguous regions while discretizing confident ones.

when no clear scalar ordering aligns with the objective, optimization tends to collapse toward nearly uniform or arbitrary scores, yielding degenerate or unstable permutations.

This behavior is reflected empirically by the trained NeuralSort results reported in Table 1, where end-to-end optimization under the unsupervised monotonicity loss fails to recover meaningful permutations beyond very small problem sizes. We emphasize that this outcome is not an implementation artifact, but a consequence of the operator-based formulation: NeuralSort and SoftSort approximate sorting, whereas our approach learns permutations as latent variables whose structure emerges only from the consistency of the reordered data.

These observations highlight a conceptual distinction rather than a deficiency of prior work. Operator-based sorting relaxations and latent permutation learning address different problem regimes, and the former is not designed to resolve heterogeneous uncertainty under weak unsupervised supervision. The same reasoning applies to SoftSort, which similarly embeds the sorting operation into the relaxation and therefore exhibits behavior analogous to that under weak unsupervised objectives.

## M. Additional Figures

## N. Details of Entropy-Adaptive Gumbel–Sinkhorn

When computing entropies for the controller, we avoid evaluating $\log 0$ by clamping the deterministic soft assignment $Q = \text{Sinkhorn}(\beta_0(t) \, S)$ entrywise:

$$\tilde{Q}_{ij} = \max(Q_{ij}, \varepsilon), \qquad \varepsilon = 10^{-8}.$$

Entropies are computed using $\tilde{Q}$. This does not affect the Sinkhorn constraints and is used only for stable entropy estimation.

In addition We clamp entrywise temperatures to $\beta_{ij} \leftarrow \text{clip}\left(\beta_{ij}, \frac{\beta_0(t)}{1+b_{\max}}, \beta_0(t)\right)$, to ensure numerical stability.

## O. Adaptive inverse temperature as local cost scaling

Let $\mathcal{B}_n$ denote the Birkhoff polytope of doubly stochastic matrices and let $C = -S$ denote costs. Ignoring Gumbel noise for clarity, Sinkhorn with global inverse temperature $\beta$ corresponds to the entropically regularized assignment problem

$$P_\beta = \arg \min_{P \in \mathcal{B}_n} \ \langle C, P \rangle + \varepsilon \sum_{i,j} P_{ij} \log P_{ij}, \quad \varepsilon = \frac{1}{\beta}.$$

Larger $\beta$ (smaller $\epsilon$) yields sharper (lower-entropy) solutions, while smaller $\beta$ encourages more diffuse assignments.

In what follows, we fix the entropy regularization weight to one and interpret inverse temperature purely as cost scaling.

Under this convention, the global inverse temperature $\beta$ corresponds to the objective $\langle \boldsymbol{C}, \boldsymbol{P} \rangle + \frac{1}{\beta} \sum_{i,j} P_{ij} \log P_{ij}$, while an entrywise inverse-temperature field $\boldsymbol{\beta}$ corresponds to locally rescaled costs $\boldsymbol{C} \mapsto \boldsymbol{\beta} \odot \boldsymbol{C}$ with a fixed entropy term.

Our adaptive formulation replaces the global inverse temperature with an entrywise field $\boldsymbol{\beta} \in \mathbb{R}_+^{n \times n}$ by applying Sinkhorn normalization to $\boldsymbol{\beta} \odot \boldsymbol{S}$ (equivalently $-\boldsymbol{\beta} \odot \boldsymbol{C}$). This corresponds to solving an entropically regularized assignment problem with locally rescaled costs:

$$\boldsymbol{P_\beta} = \arg \min_{\boldsymbol{P} \in \mathcal{B}_n} \ \langle \boldsymbol{\beta} \odot \boldsymbol{C}, \, \boldsymbol{P} \rangle + \sum_{i,j} P_{ij} \log P_{ij}. \tag{7}$$

**Proposition 2** (Adaptive inverse temperature as local cost scaling)**.** *Consider the deterministic Sinkhorn mapping applied to a score matrix $\boldsymbol{S}$ with an entrywise inverse-temperature field $\boldsymbol{\beta}$:*

$$\boldsymbol{P} = \mathrm{Sinkhorn}(\boldsymbol{\beta} \odot \boldsymbol{S})\,.$$

*Then $\boldsymbol{P}$ is the unique minimizer of the entropically regularized assignment problem* (7)*, where $\boldsymbol{C} = -\boldsymbol{S}$.*

This follows directly from the interpretation of Sinkhorn normalization as KL projection onto the Birkhoff polytope (Cuturi, 2013). A proof is provided in Appendix B.

Because feasibility is enforced by global row and column constraints, local cost rescaling induces coupled changes across the assignment and cannot be reproduced by global or separable temperature parameters.

Local inverse-temperature scaling reduces relative cost contrast between competing assignments, delaying symmetry breaking in ambiguous regions while allowing confident regions to concentrate.

# P. Additional baseline geometry: OT4P

This appendix provides additional details for the OT4P comparison, including (i) the exact unsupervised setup, (ii) a targeted hyperparameter sweep, and (iii) a brief failure analysis explaining why OT4P is near chance in our unsupervised sorting regime.

## P.1. Experimental setup (unsupervised sorting)

**Data.** Training lists are sampled i.i.d. from a uniform distribution (either $[0, 1]$ or $[10, 11]$) and randomly permuted. The ground-truth sorted order and permutation are stored for evaluation only.

**Backbone model.** We use the same lightweight Conv1D stack as in the GS experiments, with a $1 \times 1$ convolutional architecture that processes each element independently and outputs assignment scores $\boldsymbol{S} \in \mathbb{R}^{n \times n}$.

**Relaxation.** OT4P maps $\boldsymbol{S}$ to a temperature-controlled orthogonal matrix which is used as a permutation-like operator (Guo et al., 2024).

**Training objective.** We train OT4P under the same unsupervised monotonicity-violation objective used throughout: a hinge-squared penalty on negative adjacent differences after applying the predicted permutation-like matrix to the input list.

**Evaluation.** At test time, we extract a discrete permutation by row-wise $\arg \max$ of the OT4P output (with evaluation temperature set to $\tau = 0$ in our implementation) and report Kendall $\tau$ against the ground-truth permutation.

## P.2. Hyperparameter sweep (fairness)

To address concerns that OT4P performance may reflect under-tuning in weakly supervised settings, we ran a targeted sweep over the most sensitive OT4P controls while keeping the overall training protocol fixed (same data generator, objective, optimizer family, epochs, batch size, and seed protocol). Specifically, we varied: (i) the learning rate, (ii) the OT4P temperature schedule (initial/final $\tau$ and schedule form), and (iii) common numerical stabilizations for orthogonal-power parameterizations (e.g., restricting the $\tau$ range to avoid unstable intermediate regimes).

Table 10 reports the resulting Kendall $\tau$ values, aggregated over three seeds. OT4P is above chance only for very small $n$ and remains close to random-permutation performance for $n \geq 50$ under both value ranges.

**Table 10.** OT4P unsupervised sorting sweep results. All runs use the same data generator and unsupervised objective as GS-based methods; Kendall $\tau$ is reported on the test set as mean $\pm$ standard deviation over three seeds.

| Value range | $n = 5$ | $n = 10$ | $n = 50$ | $n = 100$ | $n = 200$ |
|---|---|---|---|---|---|
| $[0, 1]$ | $0.157 \pm 0.119$ | $0.202 \pm 0.013$ | $0.007 \pm 0.010$ | $0.000 \pm 0.014$ | $0.002 \pm 0.009$ |
| $[10, 11]$ | $0.068 \pm 0.074$ | $0.041 \pm 0.028$ | $0.011 \pm 0.017$ | $0.014 \pm 0.012$ | $0.002 \pm 0.008$ |

**P.3. Failure analysis: why OT4P is near chance under our unsupervised objective**

The near-chance behavior is consistent with two interacting limitations of this specific unsupervised sorting setup:

**(1) The unsupervised monotonicity objective is weak and locally defined on i.i.d. inputs.** The monotonicity-violation loss penalizes only adjacent inversions after reordering. On i.i.d. lists, many distinct score patterns induce similar expected local violation penalties, producing weak and high-variance gradients. As a result, the optimization often fails to settle on a consistent global ordering even when the relaxation sharpens.

**(2) OT4P-specific numerical sensitivity under fractional matrix powers.** In our experiments, intermediate temperatures occasionally produced complex-valued intermediates during fractional matrix powering, depending on the spectrum of the learned orthogonal matrices. Standard real-valued workarounds (e.g., discarding imaginary components) can break the intended orthogonal structure and degrade gradient quality, which is particularly harmful under weak unsupervised signals. Restricting the temperature range reduced these events but did not change the qualitative outcome in our setting.

**P.4. Interpretation**

Overall, we view OT4P's near-chance results here as evidence of a mismatch between a weak, local unsupervised sorting objective on i.i.d. inputs and an orthogonal-power relaxation. This should not be interpreted as a general negative result about OT4P: we expect OT4P to be better suited to settings with stronger supervision, richer context-aware scoring architectures (e.g., attention-based comparisons), or objectives that provide more global ranking signals.

# Q. Gumbel–Sinkhorn Operator

The Gumbel–Sinkhorn operator (Mena et al., 2018) provides a continuous, differentiable relaxation of sampling from the space of permutation matrices. Given a matrix of assignment logits $\boldsymbol{S} \in \mathbb{R}^{n \times n}$, i.i.d. Gumbel noise $\boldsymbol{g}$ with entries $g_{ij} \sim \text{Gumbel}(0, 1)$ is added to obtain a perturbed score matrix $\boldsymbol{S} + \boldsymbol{g}$. A temperature parameter $\tau > 0$ is then applied, yielding

$$\boldsymbol{Z} = \frac{\boldsymbol{S} + \boldsymbol{g}}{\tau}.$$

The Sinkhorn operator $\text{Sinkhorn}(\cdot)$ maps $\boldsymbol{Z}$ to an approximately doubly stochastic matrix by iteratively normalizing its rows and columns. In practice, this is implemented in log-space as alternating row-wise and column-wise log-softmax operations, ensuring numerical stability and differentiability. After a finite number of iterations, the resulting matrix

$$\boldsymbol{P} = \text{Sinkhorn}(\boldsymbol{Z})$$

lies in the Birkhoff polytope and can be interpreted as a soft permutation.

The temperature parameter $\tau$ controls the entropy of the resulting assignment: larger values of $\tau$ produce smoother, higher-entropy doubly stochastic matrices, while smaller values of $\tau$ sharpen the distribution and yield matrices that are increasingly close to discrete permutations. Importantly, because $\tau$ is applied after adding the Gumbel noise, it jointly scales both the logits and the noise, thereby controlling the effective stochasticity of the relaxation.

During training, $\tau$ is typically annealed from a high to a low value according to a fixed schedule, enabling stable optimization with smooth assignments in early iterations and progressively more discrete permutation estimates as training converges.

**Q.1. Additional ablations: smoothness loss and TA combination**

**Smoothness loss variants.** In jigsaw experiments, `smooth2k` enforces boundary consistency over a $2k$-pixel band rather than a single seam. For each horizontal neighbor pair, we concatenate the rightmost $k$ columns of the left tile with the

leftmost $k$ columns of the right tile and apply a total-variation penalty over the resulting strip; the same is done vertically. For $k = 1$, this reduces to a standard edge-to-edge smoothness loss.

**Temperature adaptation combination.** When entropy-adaptive temperature control is enabled, we compute normalized row and column entropies of the soft assignment. Row and column temperatures are increased when entropy exceeds a threshold. With `avg` (default), entrywise temperature is the arithmetic mean of row and column values. With `prod`, temperatures are multiplied, yielding larger boosts when both row and column are uncertain and producing softer assignments.

