# OpenReview forum: "Learning Permutation from Structure Without Supervision"
_ICML.cc/2026/Conference — ICML 2026 regular_

### Official Review · Reviewer_tJNR · 2026-03-10

**Soundness:** 3
**Presentation:** 2
**Significance:** 3
**Originality:** 3
**Overall Recommendation:** 5
**Confidence:** 3

**Summary:**

This article contributes a new surrogate problem and algorithm for the task of learning permutation matrices. Like in cited prior work, especially Mena et al. (2018), a continuous relaxation to doubly stochastic matrices is considered in a Gumbel–Sinkhorn assignment formulation of the problem. Unlike in prior work, different temperatures are introduced relative to rows and columns of the matrix, and these temperatures are adapted based on assignment uncertainty or external uncertainty estimates. Empirically, advantages are shown over fixed and adapted global temperatures, in applications to instances of unsupervised TSP and unsupervised jigsaw reconstruction with specific structured loss functions. The observed improvements are statistically significant.

**Compliance With Llm Reviewing Policy:**

Affirmed.

**Final Justification:**

I suggest to accept this article.

Moreover, I would encourage the authors to use the extra space provided for the camera-ready version in order to make the article even more accessible.

**Key Questions For Authors:**

The mentioning of the relaxation by Guo et al. (2024) makes me curious: Is something theoretical known about the relation to their relaxation?

**Limitations:**

yes

**Strengths And Weaknesses:**

Strengths:
- The problem of learning permutations is hard, important and clearly of interest to the ICML community.
- The method introduced here builds on well-established foundations, notably Mena et al. (2018). The changes compared to this and other prior work are described explicitly.
- The addressed limitation that "a single global temperature forces all assignments to sharpen or diffuse simultaneously, either freezing incorrect matches or preventing convergence in large problems", is made rigorous in the form of Proposition 1. The proof is correct and clearly written.
- The new idea of introducing separate temperatures for the rows and columns of the matrix is convincing, especially given that these temperatures can be adjusted also based on external uncertainty measurements, as discussed in Appendix B.1.
- The experimental results on the CelebA-Test set show a statistically significant advantage of the proposed formulation over the usual Gumbel–Sinkhorn assignment formulation with adaptation of a single global temperature.
- The experimental results (optimality gap and tour length) on the unsupervised TSP dataset are also an improvement; this improvement is significant for $n=500$.
- The discussion of related work is focused, demonstrating intimate knowledge of the recent related work on the problem.

Weaknesses:
- The article is not self-contained and therefore a bit difficult to read, especially for non-experts. E.g., it really helps to read Mena et al. (2018).

---

> ### Author Rebuttal · Authors · 2026-03-29
>
> We thank the reviewer for the careful reading and for highlighting both Proposition 1 and the statistically significant empirical gains. The two points you raise can be addressed directly: (i) we agree the current draft is not self-contained enough, and we can fix this by adding a compact GS primer and clearer intuition in the main text; (ii) regarding Guo et al. (2024) [1], the main theoretical relation is a contrast in geometry and regularization, not a known formal equivalence. Our contribution is therefore best understood as a principled improvement to Gumbel--Sinkhorn (GS) and related entropic optimal-transport relaxations, where heterogeneous assignment uncertainty induces a structural trade-off when a single global temperature is used.
>
> - **Self-containedness / readability (W1)** We agree that the current draft assumes too much familiarity with Mena et al. (2018) [2]. In the revision, we will make the paper more self-contained by adding a short GS primer directly in the main text.
>
>   Concretely, we will state upfront that GS relaxes a permutation matrix to a doubly stochastic matrix by applying Sinkhorn normalization to perturbed logits, and that the temperature controls the entropy of the relaxed assignment: high temperature yields diffuse, high-entropy matrices, while low temperature yields sharper, near-discrete assignments.
>
>   We will also move the main intuition earlier and narrow and make the scope more explicit. Our claim is not that a single global temperature is always inadequate. The intended claim is that when assignment uncertainty is heterogeneous, a single global temperature forces all rows and columns to sharpen or diffuse together. Some assignments may already be confident, while others remain ambiguous; lowering the global temperature can prematurely freeze incorrect matches, while keeping it high can prevent confident assignments from discretizing. Our method addresses exactly this issue by introducing row- and column-wise temperatures, allowing different parts of the assignment to evolve at different rates.
>
>   To make this easier to follow, we will revise the exposition around Proposition 1 and the adaptive variant by adding: (i) a short notation recap, (ii) a clearer explanation of the entropy--temperature link, and (iii) smoother transitions between standard GS, the limitation of global control, and the proposed adaptive controller. We will also clarify that the method is a refinement of GS rather than a replacement for the broader permutation-learning literature.
>
> - **Relation to Guo et al. (2024) / OT4P (Q1)** Our understanding is that the relation is mainly conceptual rather than a known formal equivalence. Both approaches use a temperature-like parameter to control sharpness toward permutations, but they do so through different geometries and different regularization mechanisms.
>
>   GS operates on the Birkhoff polytope through entropic regularization. In this setting, temperature is directly tied to assignment entropy. This is what enables our entropy-adaptive formulation: we can modulate sharpening locally using row/column uncertainty. It is also why Sec. 3.4 admits an optimal-transport interpretation, where adaptive temperature corresponds to local rescaling of assignment costs under global doubly stochastic constraints.
>
>   By contrast, OT4P maps parameters via the orthogonal group by matrix exponentiation. There, temperature influences the concentration toward permutation matrices, but not through the same assignment-entropy mechanism as that induced by Sinkhorn normalization. Because of this, a comparable row- or column-wise entropy decomposition is not readily available, and the specific mechanism of our method, local control of assignment entropy, does not directly transfer.
>
>   So the careful answer is: we are not aware of a theorem establishing a formal equivalence or reduction between our relaxation and OT4P. The clearest theoretical relation we can justify is therefore a contrast between geometry and regularization: our approach relies on the entropic OT/Sinkhorn structure, whereas OT4P relies on orthogonal-group geometry. This makes our adaptive controller natural for GS-like relaxations, but not immediately applicable to OT4P.
>
>   Empirically, this distinction also appears meaningful in our setting: in App. P, after a targeted sweep over learning rate, temperature schedules, and stabilization settings, OT4P remains near chance in our weakly supervised sorting setup. We do not claim this is universal; only that in our experiments, the difference does not appear to be merely a tuning issue.
>
> References
>
> [1] OT4P: Unlocking Effective Orthogonal Group Path for Permutation Relaxation (NeurIPS, 2024)
> [2] Learning Latent Permutations with Gumbel-Sinkhorn Networks (ICLR, 2018)

---

> > ### Author Rebuttal · Reviewer_tJNR · 2026-04-01
> >
> > I am convinced that the authors will succeed in making the article more accessible with the extra space provided for the camera-ready version.
> >
> > I have also read with great interest the reivew by BwmE and find their remarks very relevant. If the article is accepted, the authors should state more explicitly in the introduction that the contribution does not consist in convergence or optimality guarantees beyond standard GS.

---

> > > ### Author Response · Authors · 2026-04-03
> > >
> > > Thank you for following up and confirming that our rebuttal addressed your concerns.
> > >
> > > We appreciate your suggestion and will clarify in the introduction that the method does not provide convergence or optimality guarantees beyond standard GS.
> > >
> > > Best regards,
> > > The Authors

---

### Official Review · Reviewer_BwmE · 2026-03-10

**Soundness:** 3
**Presentation:** 3
**Significance:** 2
**Originality:** 2
**Overall Recommendation:** 4
**Confidence:** 2

**Summary:**

This paper studies unsupervised permutation learning, where a permutation is treated as a latent operator and optimized through structural losses defined on reordered data (e.g., monotonicity for sorting, boundary smoothness for jigsaw reconstruction, and tour length for TSP). The authors identify a fundamental limitation of standard Gumbel–Sinkhorn relaxations: using a single global temperature cannot simultaneously discretize confident assignments while preserving exploration in ambiguous ones. To address this, they introduce an entropy-adaptive variant of Gumbel–Sinkhorn that replaces the global temperature with a row- and column-dependent inverse-temperature field derived from assignment entropy, allowing uncertainty-aware sharpening. They further provide a theoretical analysis showing the incompatibility of global temperature control in heterogeneous settings and interpret their method as locally rescaling costs in an entropically regularized assignment problem. Empirically, the approach improves stability and permutation quality across sorting, jigsaw reconstruction, and unsupervised TSP.

**Compliance With Llm Reviewing Policy:**

Affirmed.

**Final Justification:**

The rebuttal addressed my main concerns, and I am updating my evaluation.

**Key Questions For Authors:**

1. Does the impossibility result extend beyond the specific block-structured construction, or is it mainly illustrative?

2. Does the entropy-adaptive scheme provide any formal stability or convergence advantages over heuristic temperature schedules?

3. Have you tested the method at substantially larger scales (e.g., n in the thousands), and how does the adaptive mechanism behave computationally and numerically in that regime?
4. How sensitive are results (especially for sorting and jigsaw) to the entropy threshold and boost parameters?

**Limitations:**

yes

**Strengths And Weaknesses:**

**Strengths**
* The paper provides a clear formalization of the limitation of global temperature control (Proposition 1), showing that a single inverse temperature cannot simultaneously discretize easy assignments while keeping ambiguous ones diffuse in heterogeneous settings. This strengthens the motivation beyond empirical observation.
* The entropy-adaptive temperature mechanism is conceptually grounded and linked to local cost rescaling in entropic optimal transport, providing a clean and principled interpretation rather than a purely heuristic modification.
* The method demonstrates improved stability and permutation quality across multiple tasks (sorting, jigsaw reconstruction, unsupervised TSP), with gains increasing as problem size and assignment ambiguity grow.
* The paper includes ablations, sensitivity analysis, and comparisons to alternative relaxations, and discusses limitations such as symmetry and ambiguity in objectives.

**Weaknesses**

* The contribution refines an existing relaxation mechanism rather than introducing a fundamentally new framework or enabling qualitatively new applications.
* The entropy-based controller relies on manually chosen thresholds and scaling parameters and lacks formal convergence guarantees or optimality results.
* Although improvements increase with problem size, experiments are limited to moderate scales (e.g., up to n = 500 in TSP), leaving scalability to substantially larger regimes untested.
* It remains unclear how strongly the proposed method would influence large-scale real-world systems beyond controlled research benchmarks.

---

> ### Author Rebuttal · Authors · 2026-03-29
>
> We thank the reviewer for the careful, balanced assessment. The main clarification is scope: we do not claim a fundamentally new permutation-learning framework or formal convergence guarantees. Our contribution is a principled optimization improvement within Gumbel--Sinkhorn (GS): when assignment uncertainty is heterogeneous, a single global temperature introduces a trade-off that tuning alone cannot eliminate, and entropy-adaptive control reduces this trade-off in practice. We also provide additional, larger-scale evidence at $n=1000$.
> - **Contribution and scope (W1, W4)** We agree that the method refines an existing relaxation mechanism rather than introducing a new framework, and we will revise the paper to state this more explicitly. Our claim is narrower: standard GS uses one global temperature, forcing all assignments to sharpen together. Our theory and experiments indicate that when some assignments are already confident while others remain ambiguous, this induces a structural trade-off between premature commitment and insufficient discretization. Our method replaces the global temperature with a row/column-dependent inverse-temperature field derived from assignment entropy; equivalently, it can be viewed as local cost rescaling under global doubly stochastic constraints. We will frame this as a principled optimization improvement, not as a claim about qualitatively new applications or broad real-world deployment.
> - **Structural scope of Proposition 1 (Q1)** Prop. 1 is a minimal illustrative example, not a universal impossibility theorem. Its role is to isolate the mechanism: heterogeneous assignment difficulty induces incompatible requirements on a single global temperature. More generally, when assignment margins vary substantially across rows or columns, the inverse temperature needed to discretize confident assignments tends to conflict with that needed to preserve uncertainty in ambiguous ones. Our experiments support this broader interpretation. In hard sorting, across three annealing schedules (150/100/50 epochs), global GS consistently shows the same trade-off: aggressive schedules commit too early, while conservative schedules do not sharpen enough. Entropy-adaptive GS improves Kendall-$\tau$ by $+0.122$, $+0.100$, and $+0.059$ across these schedules, and even its weakest configuration ($0.061$) matches or exceeds the best global GS result ($0.059$). We will clarify that Prop. 1 illustrates a broader structural limitation rather than proving a full impossibility result.
> - **Formal guarantees and stability claims (W2, Q2)** We agree that the current paper does not provide formal convergence or optimality guarantees beyond standard GS, and we will state this more explicitly. Our claim is empirical and mechanistic rather than a theorem of improved convergence. To isolate the effect from training dynamics and stochastic perturbations, we also evaluate a controlled synthetic setting with fixed logits and heterogeneous ambiguity across assignments. At $\tau=0.8$, entropy-adaptive control increases hard-row entropy by approximately $0.016$--$0.018$ under both deterministic Sinkhorn and row-wise softmax relaxations, while easy-row true-column mass changes by less than $0.01$. This suggests the benefit lies in resolving heterogeneous uncertainty rather than in a particular stochastic schedule or training procedure. Formal guarantees are an important direction for future work, but they are non-trivial due to the interaction among stochastic perturbations, Sinkhorn normalization, and non-convex structural objectives.
> - **Larger scales and computational behavior (W3, Q3)** We agree that evaluating beyond $n=500$ is important. In the current paper, we report up to $n=500$ (TSP), where the dominant cost is Sinkhorn normalization rather than the entropy-adaptive controller itself. The controller adds row/column entropy computation and element-wise scaling, both of which are linear in the number of assignments and negligible relative to Sinkhorn normalization. We additionally evaluated larger-scale TSP with $n=1000$. Across three random seeds, baseline UTSP achieves an average tour length of $(82.43 \pm 2.69)$, corresponding to a gap of $(77.44$% $\pm 5.79$%). Incorporating GS improves performance to $(79.41 \pm 1.39)$ (gap $(70.95$% $\pm 3.03$%)). Adding entropy-adaptive temperature control further reduces the tour length to $(77.88 \pm 1.39)$, with a gap of $(67.65$% $\pm 3.00$%). We did not observe additional numerical instability in this regime. We agree that this still does not establish behavior in the many thousands or in large-scale real-world systems, and we will calibrate the claim accordingly.

---

> > ### Author Rebuttal · Reviewer_BwmE · 2026-04-06
> >
> > My main concerns have been addressed.

---

> > > ### Author Response · Authors · 2026-04-07
> > >
> > > Dear Reviewer,
> > >
> > > Thank you for the follow-up. We’re glad the rebuttal addressed your concerns.
> > >
> > > If there are any remaining points or clarifications that would be helpful for your final assessment, we would be happy to provide them.
> > >
> > > Best regards,
> > >
> > > The Authors

---

### Official Review · Reviewer_RJ5B · 2026-03-12

**Soundness:** 3
**Presentation:** 3
**Significance:** 3
**Originality:** 3
**Overall Recommendation:** 5
**Confidence:** 2

**Summary:**

This paper addresses unsupervised permutation learning, where supervision is provided only through structural properties of the reordered output rather than ground truth permutations. The authors identify a key limitation in existing differentiable methods like Gumbel-Sinkhorn: a single global temperature cannot simultaneously discretize confident assignments while preserving exploration in ambiguous regions, leading to instability for large or heterogeneous problems. To overcome this, they propose entropy adaptive Gumbel-Sinkhorn, which assigns a row and column specific inverse temperature based on assignment entropy, allowing confident assignments to discretize early while keeping uncertain ones diffuse. This approach is theoretically grounded as a form of local cost rescaling under the Birkhoff polytope and is empirically validated across list sorting, jigsaw image reconstruction, and unsupervised TSP, showing improvde convergence stability and higher quality permutations, particularly in large scale or ambiguous settings.

**Compliance With Llm Reviewing Policy:**

Affirmed.

**Final Justification:**

The questions raised in the rebuttal have been addressed and I am satisfied. I am increasing my score to 5.

**Key Questions For Authors:**

1. How sensitive is entropy-adaptive temperature control to the choice of hyperparameters across tasks not included in the paper? Could poor choices significantly degrade performance?

2. In highly symmetric or weakly informative permutation objectives, could adaptive temperature alone suffice, or is some form of anchor or symmetry-breaking always required?

3. Could the proposed approach be generalized to partial permutations or constrained combinatorial problems beyond assignment like settings?

4. For OT4P and other non-Birkhoff relaxations, how would entropy-adaptive temperature perform if optimized more carefully? This could clarify whether the approach is broadly applicable beyond the Sinkhorn/Birkhoff setting.

**Limitations:**

Yes.

**Strengths And Weaknesses:**

Strengths

1. The method is well founded with theoretical justification for entropy-adaptive temperature and empirical validation across multiple tasks.
2. Clear improvement over existing global temperature approaches, especially for large or ambiguous permutation problems.
3. Well-motivated problem with broad applicability in unsupervised permutation learning and combinatorial optimization.
4. Experiments are thorough, covering sorting, jigsaw reconstruction, and unsupervised TSP, with consistent performance gains.
5. Presentation is generally clear, with figures and algorithms aiding comprehension.

Weaknesses

1. Introduces additional hyperparameters (entropy threshold, max boost), which may require tuning for new tasks.
2. Limited discussion on failure modes in highly symmetric or weakly informative objectives.
3. Some experiments rely on small anchor information to ensure convergence, which slightly reduces the purely unsupervised claim.
4. The method is specific to assignment like permutation problems. Applicability to more complex or structured combinatorial settings is suggested but not fully demonstrated.
5. OT4P evaluation is brief, leaving some uncertainty about generality beyond Birkhoff-based relaxations.

---

> ### Author Rebuttal · Authors · 2026-03-29
>
> We thank the reviewer for the careful, balanced assessment. The main clarification is scope: our claim is not that entropy-adaptive temperature solves identifiability or all structured combinatorial problems, but that it addresses a specific optimization bottleneck in permutation learning with heterogeneous assignment uncertainty. Within that scope, we already have concrete evidence on robustness, symmetry limitations, and generality beyond the exact GS implementation.
> - **Hyperparameter robustness (W1, Q1)** The method introduces two hyperparameters, the entropy threshold $H_0$ and maximum boost $b_{\max}$. In our experiments, performance is not highly sensitive to these choices. App. K sweeps $H_0 \in \{0.55,0.65,0.75,0.85\}$ and $b_{\max} \in \{0.05,0.10,0.20,0.35\}$ on TSP with $n=500$, and performance varies smoothly (approximately $49$--$50$) without instability. So tuning helps, but poor choices within a broad range do not significantly degrade performance.
>   We also tested robustness to the annealing schedule in a controlled sorting experiment ($n=300$, values in $[10,11]$), comparing schedules annealing $\tau=1.5 \to 0.5$ over $150$, $100$, and $50$ epochs. Standard GS is quite sensitive, whereas entropy-adaptive temperature improves performance across all schedules by $+0.122$, $+0.100$, and $+0.059$ Kendall-$\tau$, respectively. Under the most aggressive schedule, entropy-adaptive GS retains about $34$% of its performance ($0.181 \rightarrow 0.061$), whereas global GS retains about $3$% ($0.059 \rightarrow 0.002$). To rule out tuning effects, we also compare against the best global schedule: best global GS reaches Kendall $\tau=0.059$, while entropy-adaptive GS reaches $0.181$ in the same setting. We will foreground the evidence of robustness more clearly.
> - **Highly symmetric or weakly informative objectives; anchors (W2, W3, Q2)** We agree that adaptive temperature alone does not resolve fundamental ambiguity. In highly symmetric or weakly informative objectives, multiple permutations can satisfy the structural loss equally well, so some symmetry breaking is required. This is why, in jigsaw, we use a small number of anchors to make the problem well-posed while keeping the setting predominantly unsupervised.
>   This limitation comes from the objective, not the temperature design. Entropy-adaptive temperature changes when and where assignments sharpen during optimization, but not the set of valid solutions. When the objective is sufficiently informative, we do observe successful convergence without anchors; when it is not, anchors or additional structure remain necessary. We will revise the paper to state this explicitly and narrow the claim accordingly.
> - **Beyond assignment-like settings (W4, Q3)** We agree that the current formulation is most direct for assignment-like permutation problems over the Birkhoff polytope, where row/column entropy has a clear operational meaning. We therefore view extension beyond this setting as future work rather than a claim of full generality.
>   That said, the principle of adapting regularization to local uncertainty is not inherently tied to GS. The OT interpretation in Sec. 3.4 shows that our method can be viewed as local cost rescaling under global constraints, suggesting possible extensions to partial permutations, constrained routing, matching/OT problems, or other structured relaxations. To test whether the mechanism is specific to GS, we ran a controlled experiment with fixed logits and heterogeneous row ambiguity. Under both deterministic Sinkhorn and a row-wise softmax relaxation, we observe the same qualitative behavior: entropy-adaptive temperature increases entropy more in hard rows while leaving easy rows nearly unchanged. For example, at $\tau=0.8$, deterministic Sinkhorn changes hard-row entropy from $0.617$ to $0.633$ while easy-row true-column mass changes only from $0.572$ to $0.569$; row-wise softmax shows a similar pattern ($0.644 \rightarrow 0.662$ for hard-row entropy, $0.655 \rightarrow 0.649$ for easy-row mass). We also evaluated a small SBM graph-matching problem ($6$ nodes, $3$ seeds), where entropy-adaptive GS improved Kendall-$\tau$ from $0.1309$ to $0.1444$. We present these as qualitative evidence for broader applicability, not a definitive demonstration.
>
> References
>
> [1] Differentiable Top-k Classification Learning (ICML, 2022)
>
> [2] Adaptive Regularization in Point Set Matching (IJCV, 2016)
>
> [3] Unsupervised Learning for Solving the Traveling Salesman Problem (NeurIPS, 2023)
>
> [4] Unsupervised Learning Permutations for TSP Using Gumbel-Sinkhorn Operator (NeurIPS, 2023)
>
> [5] OT4P: Unlocking Effective Orthogonal Group Path for Permutation Relaxation (NeurIPS, 2024)

---

> > ### Author Rebuttal · Reviewer_RJ5B · 2026-04-03
> >
> > Thank you for the rebuttal. The questions raised have been addressed and I am satisfied. I will increase my score accordingly.

---

> > > ### Author Response · Authors · 2026-04-03
> > >
> > > Dear Reviewer,
> > >
> > > Thank you for the follow-up. We are glad the rebuttal addressed your concerns.
> > >
> > > We wanted to check whether the score was indeed updated on your side, or whether there may be a system delay.
> > >
> > > Best regards,
> > > The Authors

---

### Official Review · Reviewer_Ve6q · 2026-03-12

**Soundness:** 2
**Presentation:** 3
**Significance:** 3
**Originality:** 3
**Overall Recommendation:** 5
**Confidence:** 1

**Summary:**

This paper studies unsupervised permutation learning from structural objectives, without access to ground-truth permutations. The main technical idea is to replace the usual single global temperature in Gumbel–Sinkhorn with an entropy-adaptive local temperature, so that confident assignments can sharpen earlier while ambiguous ones remain softer longer. The paper evaluates this idea on sorting, jigsaw reconstruction, and routing-style settings, and argues that adaptive entropy control improves stability and permutation quality, especially as ambiguity and problem size increase.

**Compliance With Llm Reviewing Policy:**

Affirmed.

**Final Justification:**

The authors have a very detailed and comprehensive rebuttal to address my questions.

**Key Questions For Authors:**

The paper says the limitation of global temperature is “fundamental rather than a matter of tuning.” Could the authors clarify exactly how strong they want this claim to be?

How sensitive is the proposed method to the choice of base annealing schedule and to the particular form of the entropy-to-temperature mapping?

The OT4P result seems negative in the unsupervised sorting setting. Should readers interpret the method as mainly a Gumbel–Sinkhorn improvement, or as a broader principle for permutation relaxations?

**Limitations:**

The main limitation, in my reading, is that the method may be somewhat tied to the specific relaxation and training setup used here. I would also like a more direct discussion of sensitivity to schedules and hyperparameters, since temperature control is exactly what the paper is changing.

**Strengths And Weaknesses:**

__Strengths__

I like the high-level motivation. The paper starts from a real weakness of many differentiable permutation methods: using one global temperature for all assignments seems restrictive when some rows/columns become easy earlier than others. That intuition is simple and convincing.

The paper also has a reasonably clear contribution. The proposed entropy-adaptive Gumbel–Sinkhorn modifies the inverse temperature locally using row/column uncertainty, which feels like a meaningful extension rather than a minor tuning trick.

Empirically, the paper appears to test the method on more than one type of task, which is a plus. In the sorting setup, the claimed gains are especially in the harder high-ambiguity regime, which is exactly where the method is supposed to help.

__Weaknesses__

My main concern is that the paper’s central claim may be a bit stronger than the evidence. The paper argues that the limitation of global temperature is “fundamental rather than a matter of tuning,” and from my perspective that feels harder to establish convincingly. It seems plausible, but I am not sure the experiments fully prove such a strong statement.

I also was not fully clear on how broadly the method works beyond the main Gumbel–Sinkhorn setting. The paper mentions trying OT4P and says it did not recover meaningful permutations in the unsupervised sorting setup, which makes me wonder whether the benefit is tied fairly specifically to this relaxation family.

---

> ### Author Rebuttal · Authors · 2026-03-29
>
> We appreciate the thoughtful review. The main clarification is that our intended claim is narrower than a formal impossibility statement: the paper identifies a structural optimization trade-off arising from using a single global temperature under heterogeneous assignment uncertainty, and shows empirically that entropy-adaptive control reduces this trade-off in practice. We also agree that the paper's strongest empirical scope lies within the Gumbel--Sinkhorn (GS) family, and we will make both points explicit in the paper.
> - **Claim scope clarification (W1, Q1)** Our goal is not to claim a formal impossibility result. The intended claim is that, when some rows/columns are already confident while others remain ambiguous, a single global inverse temperature creates a practical structural trade-off: it cannot simultaneously (i) sharpen the confident assignments and (ii) preserve enough entropy in the ambiguous ones. Section 3.3 / Prop. 1 is meant to formalize this trade-off via a block-ambiguous construction; it is not a general impossibility theorem.
>   Empirically, we observe the same pattern across tasks: aggressive annealing tends to lead to premature commitment in ambiguous regions, while conservative annealing delays the discretization of easier regions. In an additional hard sorting experiment ($n=300$, values in $[10,11]$), we compared schedules annealing from $\tau=1.5$ to $\tau=0.5$ over 150, 100, and 50 epochs. Standard GS is strongly schedule-sensitive, whereas entropy-adaptive temperature consistently improves performance across all schedules, with Kendall-$\tau$ gains of $+0.122$, $+0.100$, and $+0.059$. Under the most aggressive schedule, entropy-adaptive temperature retains about $34$% of its performance, whereas global GS retains about $3$%. We will revise the wording to state this as a practical structural limitation of global temperature control, not a formal impossibility result.
> - **Robustness to schedules and hyperparameters (Q2, L2)** We agree this should be discussed more directly, since temperature control is exactly what we change. First, for the controller hyperparameters, Appendix K shows smooth variation across the tested grid of $H_0$ and $b_{\max}$, without instability or abrupt failure cases in those sweeps. Second, the schedule experiment above directly tests robustness to the global annealing schedule, and the gains persist across substantially different annealing rates.
>   Our calibrated claim is that the method still has hyperparameters, but within the tested ranges it behaves robustly, and the observed gains are unlikely to be explained solely by better tuning of a single global schedule. Conceptually, this is because entropy-adaptive temperature allows different parts of the assignment to evolve at different rates: confident assignments can sharpen early, while ambiguous regions remain diffuse.
> - **Scope beyond GS and interpretation of OT4P results (W2, Q3, L1)** The strongest empirical claim in this paper is GS-centric. GS is the natural setting because temperature directly controls the entropy of the doubly stochastic relaxation, so local entropy-based adaptation is meaningful and easy to implement. We did test OT4P in the same unsupervised sorting setup (Appendix P), with a targeted sweep over learning rate, temperature schedules, and stabilization settings. In that setting, OT4P remained near chance: mean Kendall-$\tau$ was approximately $0.019$ at $n=50$, $0.007$ at $n=100$, and $-0.002$ at $n=200$.
>   We therefore do not claim broad empirical success across all permutation relaxations. Our more qualified takeaway is that entropy-adaptive temperature is most directly applicable when the relaxation has a clear local entropy/discretization control, analogous to GS. For relaxations where temperature is not directly tied to assignment entropy, such as matrix-powering OT4P, the same controller is not guaranteed to transfer. So readers should interpret the paper primarily as an improvement to the Gumbel--Sinkhorn family, while the broader principle is that effective permutation learning can benefit from controlling where and when entropy collapses.
> - **Additional scale evidence** To address whether the effect persists at larger scale, we also observe the same qualitative benefit in additional TSP experiments with $n=1000$: UTSP gives average tour length $82.43 \pm 2.69$ (gap $77.44$% $\pm 5.79$%), GS improves to $79.41 \pm 1.39$ (gap $70.95$% $\pm 3.03$%), and entropy-adaptive temperature further improves to $77.88 \pm 1.39$ (gap $67.65$% $\pm 3.00$%). We will revise the paper to make the scope calibration explicit: the contribution is a practically useful, uncertainty-aware improvement to GS, with direct evidence on claim strength, robustness, and scope.
>
> References
>
> [1] Learning Latent Permutations with Gumbel-Sinkhorn Networks (ICLR, 2018)
>
> [2] OT4P: Unlocking Effective Orthogonal Group Path for Permutation Relaxation (NeurIPS, 2024)

---

> > ### Author Rebuttal · Reviewer_Ve6q · 2026-04-03
> >
> > Thank you for the detailed responses and for addressing my questions. I raise the score accordingly.

---

> > > ### Author Response · Authors · 2026-04-04
> > >
> > > Dear Reviewer,
> > >
> > > Thank you for the follow-up. We are glad the rebuttal addressed your concerns.
> > >
> > > Best regards,
> > >
> > > The Authors

---

### Decision · Program_Chairs · 2026-04-30

**Decision:**

Accept (regular)

**Comment:**

This paper tackles unsupervised permutation learning using Gumbel-Sinkhorn. The authors demonstrate that a single global temperature may struggle to handle heterogeneous assignment ambiguity, as it forces all assignments to sharpen or diffuse at the same rate. To resolve this, the authors introduce an entropy-adaptive Gumbel-Sinkhorn formulation that calculates a row- and column-dependent inverse-temperature field based on local uncertainty. This approach is evaluated empirically on list sorting, jigsaw reconstruction, and unsupervised Traveling Salesman Problem (TSP) tasks.

Strengths:
- The method effectively identifies and mitigates a known issue in learning differentiable permutations.
- Proposition 1 provides a formalization of the limitations of global temperature control under a specific case (block-structured construction).
- The authors scaled the TSP evaluation to $n=1000$.

Weaknesses:
- The core contribution is a heuristic applied to Gumbel-Sinkhorn, rather than a fundamentally new theoretical framework.
- The method does not offer formal convergence or optimality guarantees.
- The experimental scale remains moderate and does not demonstrate large-scale real-world deployment.

Overall, this is a weak accept.

Additional AC suggestions for the camera-ready version:

- Tone down claims regarding Proposition 1: As discussed during the rebuttal, Proposition 1 should be explicitly presented as an illustrative example demonstrating a practical optimization bottleneck, rather than a universal impossibility theorem.

- Clarify the role of Gumbel noise: The text should better explain why the Sinkhorn operator alone (for differentiable deterministic mapping) is insufficient, and why injecting Gumbel noise is useful (differentiable sampling, encouraging exploration, etc).